

# Global impacts of tropospheric halogens (Cl, Br, I) on oxidants and composition in GEOS-Chem

T. Sherwen[1], J. A. Schmidt[2], M. J. Evans[1,3], L. J. Carpenter[1], K. Großmann[4,a], S. D. Eastham[5], D. J. Jacob[5], B. Dix[6], T. K. Koenig[6,7], R. Sinreich[6], I. Ortega[6,7], R. Volkamer[6,7], A. Saiz-Lopez[8], C. Prados-Roman[8,b], A. S. Mahajan[9], and C. Ordóñez[10]

[1]Wolfson Atmospheric Chemistry Laboratories (WACL), Department of Chemistry, University of York, York, YO10 5DD, UK
[2]Department of Chemistry, Copenhagen University, Universitetsparken, DK-2100 Copenhagen O, Denmark
[3]National Centre for Atmospheric Science (NCAS), University of York, York, YO10 5DD, UK
[4]Institute of Environmental Physics, University of Heidelberg, Heidelberg, Germany
[5]School of Engineering and Applied Sciences, Harvard University, Cambridge, MA, USA
[6]Department of Chemistry and Biochemistry, University of Colorado, Boulder, CO 80309-0215, USA
[7]Cooperative Institute for Research in Environmental Sciences, University of Colorado, Boulder, CO 80309-021, USA
[8]Department of Atmospheric Chemistry and Climate, Institute of Physical Chemistry Rocasolano, CSIC, Madrid, 28006, Spain
[9]Indian Institute of Tropical Meteorology, Maharashtra, 411008, India
[10]Dpto. Física de la Tierra II, Facultad de Ciencias Físicas, Universidad Complutense de Madrid, 28040 Madrid, Spain
[a]Now at: Joint Institute For Regional Earth System Science and Engineering (JIFRESSE), University of California Los Angeles, Los Angeles, CA, 90095, USA
[b]Atmospheric Research and Instrumentation Branch, National Institute for Aerospace Technology (INTA), Madrid, Spain

*Correspondence to:* Tomás Sherwen (ts551@york.ac.uk)

**Abstract.** We present a simulation of the global composition of the troposphere which includes the chemistry of halogens (Cl, Br, I). Building on previous work within the GEOS-Chem model we include emissions of inorganic iodine from the oceans, anthropogenic and biogenic sources of halogenated gases, gas phase chemistry, and a parameterised approach to heterogeneous halogen chemistry. Consistent with Schmidt et al. (2016) we do not include sea-salt de-bromination. Observations of halogen

5    radicals (BrO, IO) are sparse but the model has some skill in reproducing these. IO shows both high and low biases in different datasets, BrO concentrations though appear to be modelled low. Comparisons to the very sparse observations dataset of reactive Cl species suggests the model represents a lower limit on impacts due to likely underestimates in emissions and therefore burdens. Inclusion of Cl, Br, I results in a general improvement in simulation of ozone ($O_3$) concentrations, except in polar regions where the model now underestimates $O_3$ concentrations. Halogen chemistry reduces the global tropospheric $O_3$ burden

10   by ~15 %, with the $O_3$ lifetime reducing from 26 days to 22 days. Global mean OH concentrations of 1.34 x$10^6$ molecules cm$^{-3}$ are 4.5 % lower than in a simulation without halogens, leading to an increase in the $CH_4$ lifetime (6.5 %) due to OH oxidation from 7.48 years to 7.96 years. Oxidation of $CH_4$ by Cl is small (~1 %) but Cl oxidation of other VOCs (ethane, acetone, and propane) can be significant (~9-18 %). Oxidation of VOCs by Br is smaller, representing 2.1% of the loss of acetaldehyde and 0.6% of the loss of formaldehyde.





# 1 Introduction

To address problems such as air quality degradation and climate change, we need to understand the composition of the tropo-
sphere and its oxidative capacity. A complicated relationship exists between key chemical families and species such as ozone
($O_3$), $HO_X$ ($HO_2$+OH), $NO_X$ ($NO_2$+NO) and organic compounds which include carbon monoxide (CO), methane ($CH_4$), hy-
drocarbons and oxygenated volatile organic compounds (VOCs) (see for example Monks et al. (2015)). The most important of
tropospheric oxidants is OH, which is itself produced indirectly through photolysis of $O_3$. Oxidants control the concentrations
of key climate and air-quality gases and aerosols (including $O_3$, methane, sulfate aerosol, and secondary organic aerosols)
(Monks et al., 2009; Prather et al., 2012; Unger et al., 2006). $O_3$ itself is not directly emitted, and it's tropospheric burden is
controlled by its sources through chemical productions from $NO_X$ and organic compounds, transport from the stratosphere,
and loss via deposition and chemical reactions (Monks et al., 2015).

Halogens (Cl, Br, I) are known to destroy $O_3$ through catalytic cycles, such as that shown in reactions 1-3 (Chameides and
Davis, 1980). Tropospheric halogens have also been shown to change OH concentrations (Bloss et al., 2005) and perturb OH
to $HO_2$ ratios towards OH (Chameides and Davis, 1980). Halogens perturb the NO to $NO_2$ ratio and reduce $NO_X$ concentra-
tions by hydrolysis of $XNO_3$. These perturbations also indirectly decrease $O_3$ formation (von Glasow et al., 2004). Halogens
directly oxidise organics species, with Cl radical reactions proceeding the fastest (Atkinson et al., 2006; Sander et al., 2011).
They also play an important role in determining the chemistry of mercury (Holmes et al., 2009; Parrella et al., 2012; Wang
et al., 2015; Coburn et al., 2016). The literature on tropospheric halogens has been the topic of several recent reviews, which
cover the background in more detail (Simpson et al., 2015; Saiz-Lopez et al., 2012b). However, many uncertainties still exist,
notably with heterogeneous halogen chemistry (Abbatt et al., 2012), and gas-phase iodine chemistry (Saiz-Lopez et al., 2014;
Sommariva and von Glasow, 2012).

$$O_3 + X \rightarrow XO + O_2 \tag{1}$$

$$HO_2 + XO \rightarrow HOX + O_2 \tag{2}$$

$$HOX + h\nu \rightarrow OH + X \tag{3}$$

Net: $HO_2 + O_3 \rightarrow 2O_2 + OH$

Tropospheric halogen chemistry has been studied in box model studies (see Simpson et al. 2015 and citations within) and
more recently in global models ( e.g. Parrella et al. 2012; Saiz-Lopez et al. 2012a, 2014; Schmidt et al. 2016; Sherwen et al.
2016). Modelling has sought to quantify emissions budgets and evaluate these on a global scale (Bell et al., 2002; Ziska et al.,
2013; Hossaini et al., 2013; Ordóñez et al., 2012). Global studies have considered impacts of halogens in the troposphere
(Parrella et al., 2012; Saiz-Lopez et al., 2012a, 2014; Schmidt et al., 2016; Sherwen et al., 2016) and reported reductions in the
tropospheric $O_3$ burden by up to ∼15 %. However, this field of research is quickly evolving, with new halogen sources such
as inorganic ocean iodine (Carpenter et al., 2013; MacDonald et al., 2014) and $ClNO_2$ produced from $N_2O_5$ hydrolysis on
sea-salt (Roberts et al., 2009; Bertram and Thornton, 2009; Sarwar et al., 2014) now appearing to be globally important.





Previous studies of halogen chemistry within the GEOS-Chem (www.geos-chem.org) model have focussed on either bromine or iodine chemistry. Parrella et al. (2012) presented a bromine scheme and its effects on oxidants in the past and present atmosphere. Eastham et al. (2014) presented the Unified tropospheric-stratospheric Chemistry eXtension (UCX), which added a stratospheric bromine and chlorine scheme. This chlorine scheme was then employed in the troposphere with an updated

heterogeneous bromine and chlorine scheme by Schmidt et al. (2016). An iodine scheme was employed in the troposphere to consider present day impacts of iodine on oxidants (Sherwen et al., 2016), which used the representation of bromine chemistry from Parrella et al. (2012). Up this point, however, the coupling of chlorine, bromine, and iodine in the GEOS-Chem model and its subsequent impact on the simulated composition of the atmosphere has not been described.

Here we present such a coupled halogen scheme within GEOS-Chem and consider tropospheric impacts of halogens. This

simulation includes recent updates to chlorine (Eastham et al., 2014; Schmidt et al., 2016), bromine (Parrella et al., 2012; Schmidt et al., 2016), and iodine (Sherwen et al., 2016) chemistry with further updates and additions described in Section 2. In Section 3 we describe the modelled distribution of inorganic halogens (Section 3.1-3.3), and compare with observations (Section 3.4). We then outline the impact on oxidants (Section 4.1-4.2), organic compounds (Section 4.3), and other species (Section 4.4).

## 2  Model Description

This work uses the GEOS-Chem chemical transport model (www.geos-chem.org, version 10) run at $4°x5°$ spatial resolution. The model is forced by assimilated meteorological and surface fields from NASA's Global Modelling and Assimilation Office (GEOS-5) . The model chemistry scheme includes $O_X$, $HO_X$, $NO_X$, and VOC chemistry as described in Mao et al. (2013). Dynamic and chemical time-step are 30 and 60 minutes, respectively. Stratospheric chemistry is modelled using a linearised

mechanism as described by Murray et al. (2012).

We update the standard model chemistry to give a representation of chlorine, bromine and iodine chemistry. We describe this version of the model as "Cl+Br+I" in this paper. It is based on the iodine chemistry described in Sherwen et al. (2016) with updates to the bromine and chlorine scheme described by Schmidt et al. (2016) and Eastham et al. (2014). We have made a range of updates beyond these. Updated or new reactions not included in Sherwen et al. (2016), Schmidt et al. (2016), or

Eastham et al. (2014) are given in Table 1 with a full description of the halogen chemistry scheme used given in Appendix Tables 6-9.

For the photolysis of $I_2O_X$ (X=2,3,4) we have adopted the absorption cross-sections reported by Gómez Martín et al. (2005) and Spietz et al. (2005) and used the $I_2O_2$ cross-section for $I_2O_4$. A quantum yield of unity was assumed for all $I_2O_X$ species. It is noted that recent work has used an unpublished spectrum for $I_2O_4$ that is much lower that $I_2O_3$ Saiz-Lopez et al. (2014),

but this is not expected to have a large effect on conclusions presented here.

The parameterisation for oceanic iodide concentration was changed from Chance et al. (2014) to MacDonald et al. (2014) as the latter resulted in an improved comparison with observations (see Section 7.5 of Sherwen et al. 2016).





The product of acid catalysed di-halogen release following I$^+$ (HOI, INO$_2$, INO$_3$) uptake was updated from I$_2$ as Sherwen et al. (2016) to yield IBr and ICl following McFiggans et al. (2002). Acidity is calculated online through titration of sea salt aerosol by uptake of sulfate dioxide (SO$_2$), nitric acid (HNO$_3$) and sulfuric acid (H$_2$SO$_4$) as described by Alexander (2005). Re-release of IX (X=Cl,Br) is only permitted to proceed if the sea salt is acidic (Alexander, 2005). Thus aerosol cycling of

IX in the model is not a net source of I$_Y$ (and may be a net sink on non-acid aerosol) but alters the speciation (Sherwen et al., 2016). The ratio between IBr and ICl was set to be 0.15:0.85 (IBr:ICl), instead of the 0.5:0.5 used previously (Saiz-Lopez et al., 2014; McFiggans et al., 2000). A ratio of 0.5:0.5 gives a large overestimate of BrO with respect to the observations used in Section 3.4.2 (Read et al., 2008; Volkamer et al., 2015). We attributed this reduction to the de-bromination of sea-salt which we do not consider here, and the potential for the model to over estimate the BrOx lifetime. This is discussed further in the next

section but future laboratory and field studies of these heterogenous process are needed to help constrain these parameters.

Iodine on aerosol is represented in the model with separate tracers based on the aerosol on which irreversible uptake occurs (see Table 8). We include 3 iodine aerosol tracers to represent iodine on accumulation and coarse mode sea-salt and on sulfate aerosol. The physical properties of the iodine aerosol tracers are assumed to be the same as its parent aerosol as previously described for sulfate (Alexander et al., 2012) and sea-salt aerosol (Jaeglé et al., 2011).

We have added to the chlorine chemistry scheme described by Eastham et al. (2014) to include more tropospheric relevant reactions based on the JPL 10-6 compilation (Sander et al., 2011) and IUPAC (Atkinson et al., 2006). The heterogenous reaction of N$_2$O$_5$ on aerosols was updated to yield products of ClNO$_2$ and HNO$_3$ (Bertram and Thornton, 2009; Roberts et al., 2009) on sea salt, and 2HNO$_3$ on other aerosol types. Reaction probabilities are unchanged (Evans and Jacob, 2005).

Deposition and photolysis of inter-halogen species (ICl, BrCl, IBr) and the reaction between ClO and IO were also included

(Sander et al., 2011).

## 3   Model results

We run the model for two years (1/1/2004 to 1/1/2006), discarding the first year as a "spin-up" period and using the second year (2005) for analysis. Non-halogen emissions are described in Sherwen et al. (2016). A reference simulation without any halogens ("NOHAL") was also performed. Where comparisons with observations are shown, the model is run for the appropriate year

with a 3 months "spin-up" before the observational dates, unless explicitly stated otherwise. The appropriate month from the 2005 simulation is used as the initialisation for these observational comparisons to account for inter-annual variations. The model is sampled at the nearest timestamp and grid box. The model only calculates chemistry in the troposphere. To avoid confusion we do not show results above the tropopause (lapse rate of temperature falls below 2 K/km).

### 3.1   Emissions

The emissions fluxes of chlorine, bromine, and iodine species are shown in Figure 1 with global totals in Table 2. We do not consider the Cl and Br contained within sea-salt as emitted in our simulation, following Schmidt et al. (2016) until a chemical





process liberates them into the gas-phase. These processes are the uptake of $N_2O_5$ on sea-salt and uptake of $I^+$ species on sea-salt. We do not include explicit sea-salt de-bromination for reasons described in Schmidt et al. (2016).

The organic iodine ($CH_3I$, $CH_2I_2$, $CH_2ICl$, $CH_2IBr$) emissions are from Ordóñez et al. (2012) as described in Sherwen et al. (2016). Inorganic iodine emissions (HOI, $I_2$) (Carpenter et al., 2013; MacDonald et al., 2014) are 28 % lower here than reported by Sherwen et al. (2016), due to use of the MacDonald et al. (2014) parameterisation for ocean surface iodide rather than that of Chance et al. (2014). Heterogeneous iodine aerosol chemistry (Section 2 and Appendix Section B1) does not lead to a net release of iodine, instead just recycling it from less active forms ($INO_2$, $INO_3$, HOI) into more active forms (ICl/IBr).

The organic bromine ($CH_3Br$, $CHBr_3$, $CH_2Br_2$) emissions have been reported previously (Parrella et al., 2012; Schmidt et al., 2016) and our simulation is consistent with this work. A further source of 0.031 Tg Br yr$^{-1}$ (3.4 % of total) is included here from $CH_2IBr$ photolysis. The heterogeneous cycling for $Br_Y$ (defined in footnote below[1]) has been updated here from Schmidt et al. (2016), as described in Section 2/Appendix B1. An additional $Br_Y$ source not considered by Schmidt et al. (2016) is iodine activated IBr release from sea salt, which amounts to 0.31 Tg Br yr$^{-1}$ and the majority (67 %) of this is tropical (22°N-22°S). With all these updates, the tropospheric mean daytime (07:00-19:00) BrO concentration is 1.1 pmol mol$^{-1}$ (0.64 pmol mol$^{-1}$ 24 hr average), which is 13 % higher than reported in Schmidt et al. (2016).

The organic chlorine emission ($CH_3Cl$, $CHCl_3$, $CH_2Cl_2$) for this simulation (Table 2) has been described previously Schmidt et al. (2016) and set using fixed surface concentrations. An additional source of 0.046 Tg Cl yr$^{-1}$(0.94 % of total) is present from $CH_2ICl$ photolysis (Sherwen et al., 2016). $ClNO_2$ production from the heterogeneous uptake of $N_2O_5$ provides a source of 0.66 Tg Cl yr$^{-1}$ (14 % of total) with the vast majority (95 %) being in the northern hemisphere, with strongest sources in coastal regions north of 20$^o$N. For June we calculate a global source of 21 Gg Cl month$^{-1}$ which is substantially less than the 62 Gg Cl month$^{-1}$ (Pers. com. Sarwar Golam 2016) calculated in a previous study (Sarwar et al., 2014). The difference in $NO_X$ concentrations due to differences in model resolution probably contributes to this. Uptake of HOI, $INO_2$ and $INO_3$ to sea-salt aerosol leads to the emission of ICl, giving an additional source of 0.78 Tg Cl yr$^{-1}$ (17.6 % of total) mostly (67 %) in tropical (22°N-22°S) locations.

Most of the emissions of Br and I species in our simulation occur in the tropics. It is notable that the chlorine emissions are more widely distributed. This is as a result of longer lifetimes of chlorine precursor gases which moves their destruction further from their emissions and that the $ClNO_2$ source is primarily in the northern extra tropics.

## 3.2 Deposition of halogens

Figure 2 shows the global annual integrated wet and dry deposition of inorganic $X_Y$ (X=Cl, Br,I). Much of the deposition of the halogens occurs over the oceans (69 %, 83 %, and 90 % for $Cl_Y$, $Br_Y$ and $I_Y$ respectively). It is high over regions of significant tropical precipitation (ITCZ, Maritime continents, Indian Ocean) and much lower at the poles reflecting lower precipitation and emissions.

We find that the the major $Cl_Y$ depositional sink is HCl (85 %), with HOCl contributing 11 % and $ClNO_3$ 3.2 %. The $Br_Y$ sink is split between HBr, HOBr and $BrNO_3$ with fractional contributions of 38, 30 and 24 % respectively. The major $I_Y$ sink

---

[1]Here $X_Y$ (X=Cl,Br,I) is the sum of gas-phase inorganic species of a given halogen in units of that halogen



is HOI deposition which represents 59 % of the depositional flux. The two next largest sinks are deposition of $INO_3$ and iodine aerosol (22 % and 15 %).

## 3.3 Halogen species concentrations

Figure 3 shows the surface and zonal concentration of annual mean $I_Y$, $Br_Y$, $Cl_Y$, with Figure 4 showing the same for IO, BrO

and Cl, key halogen compounds in the atmosphere. Figure 5 showing the global molecule weighted mean vertical profile of the halogen speciation.

Inorganic iodine concentrations are highest in the tropical marine boundary layer consistent with their dominant emissions regions. The highest concentrations are calculated in the coastal tropical regions, where enhanced $O_3$ concentrations from industrial areas flow over high predicted oceanic iodide concentrations and lead to increased oceanic inorganic iodine emissions.

Within the vertical there is an average of $\sim$0.5-1 pmol mol$^{-1}$ of $I_Y$ consistent with previous model studies (Saiz-Lopez et al., 2014; Sherwen et al., 2016). The lowest concentrations of $I_Y$ are seen just above the marine boundary layer where $I_Y$ loss via wet deposition is most favourable due to partitioning towards water soluble HOI. At higher altitudes, lower temperature and high photolysis rates push the $I_Y$ speciation to less water soluble compounds (IO, $INO_3$) and hence the $I_Y$ lifetime is longer. IO concentrations (Figure 4) follow the concentrations of $I_y$ with high concentrations in the tropical marine boundary layer. The

IO concentration increases into the upper troposphere reflecting a partitioning of $I_y$ in this region towards IO (and $IONO_2$) and away from HOI. The global mean tropospheric lifetimes of $I_Y$ and $IO_X$ are 2.3 days and 1.3 minutes, respectively.

Total reactive bromine is more equally spread through the atmosphere than iodine. This reflects the longer lifetime of source species with respect to photolysis which gives a more significant source higher in the atmosphere. The highest concentrations are still found in the tropics. Unlike $I_Y$, $Br_Y$ increases significantly with altitude, with $BrNO_3$ and HOBr being the two most

20 dominant species. BrO concentrations (Figure 4) follows the concentration of inorganic bromine. In the boundary layer the highest concentrations are found in the tropical marine boundary layer concentrations are in the tropical marine boundary. BrO and IO do not strongly correlate in the tropical marine boundary layer reflecting their differing sources. BrO concentrations increase towards the upper troposphere associated with the increase in total $Br_y$. The global annual average (molecule weighted) tropospheric BrO mixing ratio in our simulation is 0.64 pmol mol$^{-1}$ (Bry=4.5 pmol mol$^{-1}$). When previous implementations

(Parrella et al., 2012; Schmidt et al., 2016) are run for the same year and model version as this work (GEOS-Chem v10), the modelled BrO concentrations are found to be 12 % lower than Schmidt et al. (2016), but 17 % higher than Parrella et al. (2012). We calculate a tropospheric lifetime of $Br_Y$ of 17 days and a $BrO_X$ lifetime of 15 minutes.

Total inorganic chlorine has a highly non-uniform distribution at the surface reflecting the dominance of the $ClNO_2$ source from $N_2O_5$ uptake on sea-salt. At the surface $ClNO_2$, HCl, BrCl and HOCl represent around 25 % of the total $Cl_Y$ each. Away

from the surface the $ClNO_2$ concentrations drop off rapidly due to the short lifetime of sea salt. HCl concentrations increase significantly into the middle and upper troposphere and dominate the $Cl_Y$ distribution. This suggests that stratospheric chlorine freed from CFCs and organic chlorine strongly contributes to free tropospheric concentrations of $Cl_Y$. However modelled $Cl_Y$ is likely a lower limit on the concentrations in the uppermost troposphere (Froidevaux et al., 2008). Cl mixing ratios are very low 0.075 fmol mol$^{-1}$ or 2000 cm$^{-3}$ in the marine boundary layer. Reactive Cl (ie not HCl) drop from the surface to around





10km where it then increases again towards to stratosphere. Cl shows a wider distrbution than IO and BrO reflecting the source wider distribution of $Cl_y$. We calculate a tropospheric lifetime of $Cl_Y$ of 15 days, a $ClO_X$ lifetime of 2 seconds, and a global tropospheric mean inorganic chlorine ($Cl_Y$) concentration of 70 pmol mol$^{-1}$ in our simulation.

The chemistry of halogens and sea-salt is highly uncertain (Simpson et al., 2015; Saiz-Lopez et al., 2012b; Abbatt et al., 2012). Estimates for sea-salt de-bromination range from 0.51 Tg yr$^{-1}$ (Parrella et al. 2012 implemented in GEOS-Chem v10 and v9-2) to 2.9 Tg yr$^{-1}$ (Fernandez et al., 2014). Some studies have also not included sea-salt de-bromination (von Glasow et al., 2004; Schmidt et al., 2016) as we do not in this work. Arguably this work therefore provides a lower estimate of bromine and chlorine sources in the troposphere.

Figure 6 shows column integrated BrO and IO, which are the major halogen species for which we have observations (see
Section 3.4). Tropospheric ClO concentrations within the troposphere are small (see Figure 5) and are therefore not shown in Fig 6. Tropical maxima are seen for both BrO and IO, with BrO concentrations decreasing towards the equator. For IO a localised maximum is seen in the Arabian Sea. The IO maximum in Antarctica reported from satellite retrievals (Schönhardt et al., 2008) is not reproduced in our model potentially reflecting the lack of polar specific processes in the model.

### 3.4   Comparison with halogen observations

The observational dataset of tropospheric halogen compounds is sparse. Previous studies that this work is based on have shown comparisons for the oceanic precursors for chlorine (Eastham et al., 2014; Schmidt et al., 2016), bromine (Parrella et al., 2012; Schmidt et al., 2016), and iodine (Bell et al., 2002; Sherwen et al., 2016; Ordóñez et al., 2012). The model performance in simulating these compounds has not changed since these previous publications so we focus here on the available observations of concentrations of IO, BrO, and some inorganic chlorine species ($ClNO_3$, HCl and $Cl_2$).

### 3.4.1   Iodine monoxide (IO)

A comparison of IO to a suite of recent remote surface observations is shown in Fig 7. The model shows an overall negative bias of 21 %. This compares with the 90 % positive bias previously reported in (Sherwen et al., 2016). This reduction in bias is due to the use of the MacDonald et al. (2014) iodide parameterisation over that of Chance et al. (2014) which has reduced the inorganic emission of iodine, along with the restriction of iodine recycling to acidic aerosol.

Figure 8 shows a comparison between modelled IO with altitude against observations in the eastern Pacific (Volkamer et al., 2015; Wang et al., 2015). In general, the model agreement with observations is good. There is an average bias of +40 % in the free troposphere (350 hPa<p< 900 hPa), which increases to +58 % in the upper troposphere (350 hPa>p> tropopause). As with the surface measurements, the model bias when comparing to IO observations (Volkamer et al., 2015; Wang et al., 2015) in the free and upper troposphere is decreased from previously reported positive biases of 73 % and 96 %, respectively
(Sherwen et al., 2016).



### 3.4.2 Bromine monoxide (BrO)

Comparisons of BrO against seasonal satellite tropospheric BrO observations from GOME-2 (Theys et al., 2011) are shown in Figure 9. As shown previously (Parrella et al., 2012; Schmidt et al., 2016) the model has some skill in capturing both the latitudinal and monthly variations in tropospheric BrO columns. However it underestimates the column BrO in the lower southern latitudes (60°S-90°S), and to a smaller degree also in lower northern latitudes (60°N-90°N) which may reflect the lack of bromine from polar (blown snow, frost flowers etc.) sources and sea-salt de-bromination processes.

Figure 11 shows modelled vertical BrO concentrations against observations in the eastern Pacific (Volkamer et al., 2015; Wang et al., 2015). We find a reasonable agreement within the free troposphere (350 hPa$<$p$<$ 900 hPa) in both the tropics and subtropics, with an average negative bias of 15 and 34 %, respectively. A similar comparisons is seen in the upper troposphere (350 hPa$>$p$>$ tropopause) show similar negative biases for the tropics and subtropics, of 20 and 24 %, respectively. The decrease in agreement seen in the TORERO comparison (Fig. 11) relative to that previously presented in Schmidt et al. (2016) is due to reduced BrCl and BrO production from slower cloud multiphase chemistry (see Sections B1-B3). We model hihjer BrO concentrations in the tropical marine boundary layer above those observed (Volkamer et al., 2015). Our modelled concentrations are lower than those reported previously (Miyazaki et al., 2016; Long et al., 2014; Pszenny et al., 2004; Keene et al., 2009).

As shown in Fig. 10, comparisons between the model and observations of BrO made at Cape Verde (Read et al., 2008; Mahajan et al., 2010) show a negative bias of 50 %. We attribute this to the high local sea-salt loadings at this site (Carpenter et al., 2010), which is situated in the surf zone. This may locally increase the BrO concentrations. The model concentrations of $\sim$1 pmol mol$^{-1}$ are however consistent with other ship borne observations made in the region (Leser et al., 2003).

Our model does not include sea-salt de-bromination and yet calculated roughly the correct concentrations of BrO. Inclusion of sea-salt de-bromination leads to excessively high BrO concentration in the model (Schmidt et al., 2016). Sea-salt de-bromination is well observed, thus the success of the model despite the lack of inclusion of this process suggest model failure in other areas. The BrO$_X$ lifetime may be too long. This is dominate by the reaction between Br and organics to produce HBr. Oceanic sources of VOCs such as acetaldehyde have been proposed (Millet et al., 2010; Volkamer et al., 2015) and a significant increase in the concentration of these species would lead to lower BrO$_X$ concentrations. Alternatively, a reduction in the efficiency of cycling of Br$_Y$ through aerosol would also have a similar effect. The aerosol phase chemistry is complex and the parameterisations used here may be too simple or fail to capture key processes (e.g. pH, organics). These all require further study in order to help reconcile the rapidly growing body of observation of both gas and aerosol phase bromine in the atmosphere with models.

### 3.4.3 Nitryl chloride (ClNO$_2$), hydrochloric acid (HCl), hypochlorous acid (HOCl) and molecular chlorine (Cl$_2$)

Very few constraints on the concentration of tropospheric chlorine species are available.

An increasing number of ClNO$_2$ observations are available (Table 3). We find that the model does reasonably well in coastal regions, but does not reproduce observations in continental regions or regions with very high NO$_X$.



Lawler et al. (2011) reports measurements of HOCl and $Cl_2$ at Cape Verde for a week in June 2009. For the first 4 days of the campaign, HOCl concentrations were higher and peaked at $\sim$100 pmol mol$^{-1}$ with $Cl_2$ concentrations peaking at $\sim$30 pmol mol$^{-1}$. For the later days, HOCl concentrations dropped to around 20 pmol mol$^{-1}$ and $Cl_2$ concentrations to $\sim$0-10 pmol mol$^{-1}$. We calculate much lower concentrations of $Cl_2$ ( $1\times10^{-3}$ pmol mol$^{-1}$) and slightly lower HOCl ( 10 pmol mol$^{-1}$) throughout the same days of the year in our analysis year (2005). This is similar to findings of Long et al. (2014), who also found better comparisons with the cleaner period of observations. Similar to the comparison with observed $ClNO_2$, our simulation underestimates HOCl and $Cl_2$.

The model does not include many sources of reactive chlorine. The failure to reproduce continental $ClNO_2$ is likely due to a lack of representation of sources such as salt plains, direct emission from power station and swimming pools, and HCl acid displacement. The inability to reproduce the very high $ClNO_2$ found in cities (Pasadena) and industrialised regions(Texas) may be due to the coarse resolution of the model compared to the spatial inhomogeneity of these observations. The failure to reproduce the Cape Verde observations may be due to the very simple aerosol phase chlorine chemistry included in the model. Overall we suggest that the model provides a lower limit estimate of the chlorine emissions and therefore burdens within the troposphere, but constraints at the surface concentrations are limited and vertical profiles are not available. Further laboratory work to better define aerosol processes and observations will be necessary to investigate the role of chlorine on tropospheric chemistry.

## 4 Impact of halogens

We now investigate the impact of the halogen chemistry on the composition of the troposphere. We start with $O_3$ and OH and then move onto other components of the troposphere.

### 4.1 Ozone ($O_3$)

Figure 12 shows changes in column, surface and zonal $O_3$ both in absolute and fractional terms between simulations with and without halogen emissions ("Cl+Br+I" vs "NOHAL"). Globally the mass-weighted, annual-average mixing ratio is reduced by 7.4 pmol mol$^{-1}$ (14.6%) with the inclusion of halogens ("Cl+Br+I"-"NOHAL")/"NOHAL"*100). A much larger percentage decrease of 25.0 % (7.2 pmol mol$^{-1}$) is seen over the ocean surface. Large percentage losses are seen in the oceanic southern hemisphere as reported previously (Long et al., 2014; Schmidt et al., 2016; Sherwen et al., 2016) reflecting the significant ocean-atmosphere exchange in this regions. The majority (65 %) of the change in $O_3$ mass due to halogens occurs in the free troposphere (350 hPa<p<900 hPa).

Comparisons of the model and observed surface and sonde $O_3$ concentrations are given in Figures 13 and 14. In the tropics the fidelity of the simulation improves with the inclusion of halogens (Schmidt et al., 2016; Sherwen et al., 2016). Sonde and surface comparisons north of $\sim$50°N and south of $\sim$60°S however show that the model now underestimates $O_3$.



The global odd oxygen budget ($O_X$, as defined in the footnote below[2]) in the troposphere with ("Cl+Br+I") and without halogens ("NOHAL") is shown in Table 4. The $O_X$ loss through chlorine, bromine, and iodine represents 0.46, 5.8 and 12 % of the total $O_X$ loss respectively, thus halogens constitute 18.2 % of the overall $O_3$ loss. The sum of halogen driven $O_X$ loss is 900 Tg $O_X$ yr$^{-1}$ , which is similar to the magnitude of loss via reaction of $O_3$ with $HO_2$ of ∼1100 Tg $O_X$ yr$^{-1}$ (23 % of total).

Halogen cross-over reactions (BrO+IO, BrO+ClO, IO+ClO) contribute little to the overall $O_3$ loss. This number compares with ∼930 Tg $O_X$ yr$^{-1}$ reported in GEOS-Chem previously by Sherwen et al. (2016). Saiz-Lopez et al. (2014) found that, between 50°S-50°N and over ocean only, halogens are responsible for the loss of 640 Tg $O_X$ yr$^{-1}$. We find a comparable value of 670 $O_X$ yr$^{-1}$ with our model.

The majority of the halogen driven $O_3$ loss (58.1 %) occurs in the free troposphere (350 hPa<p<900 hPa). Halogens represent 34.9 and 31.0 % of $O_X$ loss in the upper troposphere (350 hPa>p> tropopause) and marine boundary layer (900 hPa<p ) respectively as shown in Figure 15. The marine boundary layer $O_X$ loss attributable to halogens is equal to the 31 % reported by Prados-Roman et al. (2015a) previously, and it is slightly higher than that reported solely for iodine of 26 % (Sherwen et al., 2016).

Although the partitioning between the $O_X$ loss processes is significantly different between the simulations with halogens and without (Table 4), the overall annual $O_X$ loss only increases by 2.2 % (4933 vs 4829 Tg yr$^{-1}$). The $O_X$ production term decreases by 1.0 %. This decrease is due to a reduction in $NO_X$ concentrations due to hydrolysis of $XNO_3$ (X=Cl, Br, I). Our tropospheric $NO_X$ burden decreases by 1.7 % to 168 Gg N (see table 10) on inclusion of halogens consistent with observations and previous model studies (Long et al., 2014; von Glasow et al., 2004; Parrella et al., 2012; Schmidt et al., 2016). Globally $NO_X$ loss through $ClNO_3$ and $BrNO_3$ hydrolysis is approximately equal (1:0.86), and overall proceeds at a rate of ∼10 % of the $NO_X$ loss through the $NO_2$+OH pathway. Iodine nitrite and nitrate ($INO_2$, $INO_3$) hydrolysis is much less significant (∼0.25 % of rate of $NO_2$+OH). Net $O_X$ is the difference between the production and loss terms and the change here is much greater leading to an overall decrease in net production of tropospheric $O_3$ ($PO_X$-$LO_X$) of 26 % (159 Tg yr$^{-1}$), and a resultant in decrease $O_3$ lifetime of 14 %.

## 4.2 HO$_X$ (OH+HO$_2$)

We find that global molecule weighted average $HO_X$ (OH+$HO_2$) concentrations are reduced by 8.5 % with the inclusion of halogens, with OH decreasing by 4.5 % from 1.40x10$^6$ to 1.34x10$^6$ molecules cm$^{-3}$. Lower $O_3$ concentrations decrease the primary OH source ($O_3 \xrightarrow{h\nu} 2OH$) by 15.5 %, and the secondary OH source from $HO_2$+NO by 2.2 %.

The reduction in the sources of OH is buffered by an additional OH source from the photolysis of HOX (X=Cl, Br, I) which acts to increase the conversion of $HO_2$ to OH. Previously, Sherwen et al. (2016) showed an increase of 1.8 % in global OH concentrations on inclusion of iodine. However, increased $Br_Y$ and reduced $I_Y$ concentrations in the simulations described here mean that the increased OH source from HOX photolysis does not compensate fully for the reduced primary source, resulting

---

[2]Here $O_X$ is defined as $O_3 + NO_2 + 2NO_3 + PAN + PMN + PPN + HNO_4 + 3N_2O_5 + HNO_3 + MPN + XO + HOX + XNO_2 + 2XNO_3 + 2OIO + 2I_2O_2 + 3I_2O_3 + 4I_2O_4 + 2Cl_2O_2 + 2OClO$, where X=Cl, Br, I; PAN = peroxyacetyl nitrate; PPN = peroxypropionyl nitrate; MPN = methyl peroxy nitrate; and PMN = peroxymethacryloyl nitrate.





in an overall 4.5 % reduction in global mean OH. This buffering contributes to a smaller change in OH than report previously by Schmidt et al. (2016) of 11 %. As reported previously (Long et al., 2014; Schmidt et al., 2016), we also find the net effect of halogens on the OH:$HO_2$ ratio is a small increase (4.4 %).

### 4.3 Organic Compounds

The oxidation of volatile organic compounds (VOCs) by halogens is included in this simulation (see Table 6 for reactions). The global fractional loss due to OH, Cl, Br, $NO_3$, and photolysis for a range of organics is shown in Figure 16.

Globally, Br oxidation is small in our simulation and contributes 2.0 % to the loss of acetaldehyde ($CH_3CHO$), 0.6 % of the loss of formaldehyde ($CH_2O$), 0.26 % of the loss of $\geqslant$C4 alkenes, and $< 0.001$ % of the loss of other compounds. Recent work has suggests a significant source of oceanic oxygenated VOCs (Millet et al., 2010; Coburn et al., 2014; Sinreich et al., 2010; Mahajan et al., 2014; Lawson et al., 2015; Volkamer et al., 2015; Myriokefalitakis et al., 2008) which we do not include in this simulation. Furthermore although our modelled $Br_Y$ is broadly comparable to some previous work (Schmidt et al., 2016; Parrella et al., 2012), it is lower in the marine boundary layer than in other recent work (Long et al., 2014). The combination of these two factors suggest that our model provides a lower bounds of impacts of bromine on VOCs. Significantly higher concentrations of oVOC would decrease the BrO concentrations in the model and might then allow an increased sea-salt source of reactive bromine.

The oxidation of Volatile Organic Compounds (VOCs) by chlorine is more significant. In our simulation chlorine accounts for 18, 9, and 9 % of the global loss of ethane ($C_2H_6$), propane ($C_3H_8$), and acetone ($CH_3C(O)CH_3$)), respectively. Loss of other VOCs is globally small. This increased loss due to Cl is to some extent compensated for by the reduction in the OH concentrations that we calculate. Thus the overall lifetime of ethane, propane, and acetone changes from 131, 38, 85 days in the simulation without halogens to 120, 37, 82 in the simulation with halogens. Notably the ethane lifetime without halogens is 10% longer than it is with. Given that we consider the chlorine in the model to be a lower limit, ethane oxidation by chlorine may in reality be more significant than found here.

Methane is a significant climate gas, as it has the second highest forcing amongst well-mixed greenhouse gases from preindustrial to present day (Myhre et al., 2013). In our simulation without halogens we calculate a tropospheric chemical lifetime due to OH of 7.48 years. With the inclusion of halogen chemistry the OH concentration drops, extending the methane lifetime due to OH of become to 7.96 years (an increase of 6.5 %). However, in our halogen simulations, chlorine radicals also oxidise methane ($\sim$1 % of the total loss) shortening the lifetime to 7.89 years (0.85 %). As noted previously, the model's chlorine concentrations appear to be underestimated. Allan et al. (2007) estimate a 25 Tg $yr^{-1}$ sink for methane from Cl ($\sim$4 %), significantly higher than our estimate. Overall the model's $CH_4$ lifetime still appears to be short compared to the observationally based estimation of $9.1 \pm 0.9$ from Prather et al. (2012), but halogens decrease this bias.

In our simulations, halogens (essentially chlorine) have a significant but not overwhelming role in the concentrations of hydrocarbons (from $\sim$1 % of methane loss to $\sim$18 % of ethane loss). However, as discussed earlier the low biases seen with the very limited observational dataset of chlorine compounds would suggest that the impacts calculated here are probably lower limits.





## 4.4 Other species

With the inclusion of halogens in the troposphere there are a large number of changes in the composition of the troposphere. Figure 17 illustrates the fractional global change in burden by species (for abbreviation see footnote[3]). The spatial and zonal distribution of these changes by species family ($HO_X$, $NO_X$, $SO_X$ as defined in footnote[4]) are shown in Figure 18 and for a few

VOCs ($C_3H_8$, $C_2H_6$, acetone, and $\geqslant$C4 alkanes) in Figure 19. A tabulated form of these changes is given within the Appendix (Table 10)

As discussed in section 4.1 and 4.1, a clear decrease in oxidants ($O_3$, OH, $HO_2$, $H_2O_2$) is seen. This drives an increase in the concentrations of some VOCs (2.1 % on a per carbon basis), including CO (2.8 %) and Isoprene (3.4 %). However, as discussed, it also adds an additional Cl sink term which leads to an overall decrease in some species (e.g. $C_2H_6$, $(CH_3)_2CO$,

$C_3H_8$) particularly in the northern hemisphere oceanic regions. The $SO_X$ burden increases slightly (0.7 %), which can be attributed to decreases in oxidants.

## 5 Summary and Conclusions

We have presented a model of tropospheric composition which has attempted to include the major routes of halogen chemistry impacts. Assessment of the model performance is limited as observations of halogen species are extremely sparse. However,

given the available observations we conclude that the model has some useful skill in predicting the concentration of iodine and bromine species and appears to underestimate the concentrations of chlorine species.

Consistent with previous studies, our model shows significant halogen driven changes in the concentrations of oxidants. The tropospheric $O_3$ burden and global mean OH decreased by 14.6 %, and 4.5 % respectively, on inclusion of halogens. The methane lifetime increases by 6.5 %, improving agreement with observations.

There are a range of changes in the concentrations of other species. Direct reaction with Cl atoms leads to enhanced oxidation of hydrocarbons with ethane showing a significant response. Given the model appears to provide a lower limit for atomic Cl concentrations this suggests a major missing oxidation pathway for ethane which is currently not considered. $NO_X$ concentrations are reduced by aerosol hydrolysis of the halogen nitrates which leads to reduced global $O_3$ production. Our simulation of BrO appears to be relatively consistent with those observed, however we do not include sea-salt de-bromination mechanism.

This would suggest that either the cycling of bromine in our model is generally too fast, or that we do not have sufficiently large $BrO_X$ sinks (potentially oVOCs). Both hypothesis warrant further research.

Significant uncertainties however remain in our understanding of halogens in the troposphere. The gas phase chemistry and photolysis parameters of iodine compounds are uncertain, together with the emissions of their organic and inorganic precursors (Sherwen et al., 2016). For chlorine, bromine and iodine heterogeneous chemistry, little experimental data exists and suitable

---

[3]Abbreviated species names are defined in the GEOS-Chem manual (http://acmg.seas.harvard.edu/geos/doc/man/appendix_6.html) and here: MOH=Methanol, EOH=Ethanol, ALD2=Acetaldehyde, ISOP=Isoprene, ALK4=$\geq$C4 alkanes, $CH_3O_2$,=Methylperoxy radical, A3O2= primary $RO_2$ from $C_3H_8$, B3O2=secondary $RO_2$ from $C_3H_8$, ATO2=$RO_2$ from Acetone, R4O2=$RO_2$ from $\geq$C4 alkanes, RIO2=$RO_2$ from Acetone

[4] Here we define families of $HO_X$, $NO_X$ and $SO_X$ as follows. $HO_X$: OH + $HO_2$, $NO_X$: NO+$NO_2$, $SO_X$ : $SO_2$ + $SO_4$ + $SO_4$ on sea salt.



parameterisations for the complex aerosols found in the atmosphere are unavailable (Abbatt et al., 2012; Saiz-Lopez et al., 2012b; Simpson et al., 2015).

Understanding fully the impact of halogens on tropospheric composition will require significant development of new experimental techniques and more field observations, new laboratory studies and models which are able to exploit these developments.

## 5 Appendix A: Tabulated Burden Changes on inclusion of halogens

Table 10 gives the burdens with and without halogens and the fractional change.

## Appendix B: Gas phase Chemistry Scheme

Here is described the full halogen chemistry scheme as presented in previous work (Bell et al., 2002; Eastham et al., 2014; Parrella et al., 2012; Schmidt et al., 2016; Sherwen et al., 2016) and with updates as detailed in section 2 and Table 1. The 10 complete gas phase photolysis, bimolecular and termolecular reactions are described in Tables 5 ,6 and 7 .

### B1 Heterogenous reactions

The halogen multiphase chemistry mechanism is based on the iodine mechanism ("Br+I") described in Sherwen et al. (2016) and the coupled mechanism of Schmidt et al. (2016). The heterogenous reactions in the scheme are shown in Table 8 and with further detail individual detail on certain reactions below. The loss rate of a molecule X due to multiphase processing on 15 aerosol is calculated following Jacob (2000).

$$\frac{dn_X}{dt} = -\left(\frac{r}{D_g} + \frac{4}{c\gamma}\right)^{-1} An_X, \tag{B1}$$

where $r$ is the aerosol effective radius, $D_g$ is the gas phase diffusion coefficient of X, $c$ is the average thermal velocity of X, $\gamma$ is the reactive uptake coefficient, $A$ is the aerosol surface area concentration, and $n_X$ is the gas phase concentration of X.

### B2 Aerosols

20 We consider halogen reactions on sulfate aerosols, sea salt aerosols, and liquid and ice cloud droplets. The implementation of sulfate type aerosols in GEOS-Chem is described by Park et al. (2004) and Pye et al. (2009). Sulfate aerosols are assumed to be acidic with pH=0.

The GEOS-Chem sea salt aerosol simulation is as described by Jaeglé et al. (2011). The transport and deposition of sea salt bromide follows that of the parent aerosol. Oxidation of bromide on sea-salt produces volatile forms of bromine that are 25 released to the gas phase. Sea salt aerosol is emitted alkaline, but the alkalinity can be titrated in GEOS-Chem by uptake of $HNO_3$, $SO_2$, $H_2SO_4$ (Alexander, 2005). Sea salt aerosol with no remaining alkalinity is assumed to have pH=5. We assume no halide oxidation on alkaline sea salt aerosol.



The liquid cloud droplet surface area is modelled using cloud liquid water content from GEOS-FP (Lucchesi, 2013) and assuming effective cloud droplet radii of 10 μm and 6 μm for marine and continental clouds, respectively. The ice cloud droplet surface area is modelled in a similar manner assuming effective ice droplet radii of 75 μm. We assume that ice cloud chemistry is confined to an unfrozen overlayer surrounding the ice crystal, see Schmidt et al. (2016) for details. Cloud water
pH (typically between 4 and 6) is calculated locally in GEOS-Chem following (Alexander et al., 2012).

The reactive uptake coefficients depend on the aerosol halide concentration. For sea salt aerosol, the bromide concentration is calculated directly from the bromide content and the aerosol mass. Sea salt aerosol chloride is assumed to be in excess (see below). For clouds and sulfate aerosol, the bromide and chloride concentration is estimated assuming equilibrium between gas phase HX and aerosol phase $X^-$.

**B3   Reactive uptake coefficients**

**B3.1   $HOBr + Cl^-/Br^-$**

The reactive uptake coefficient is calculated as

$$\gamma = \left(\Gamma^{-1} + \alpha^{-1}\right)^{-1},\tag{B2}$$

where the mass accomodation coefficient for HOBr is $\alpha = 0.6$ and

$$\Gamma = \frac{4H_{\text{HOBr}}RTk_{\text{HOBr}+\text{X}^-}[\text{X}^-][\text{H}^+]l_r f(r,l_r)}{c},\tag{B3}$$

with $k_{\text{HOBr}+\text{Cl}^-} = 5.9 \times 10^9\,\text{M}^{-2}\text{s}^{-1}$ and $k_{\text{HOBr}+\text{Br}^-} = 1.6 \times 10^{10}\,\text{M}^{-2}\text{s}^{-1}$. In the equation above $c$ is the average thermal velocity of HOBr, and $f(l_r, r)$ is a reacto-diffusive correction factor,

$$f(l_r, r) = \coth\left(\frac{r}{l_r}\right) - \frac{l_r}{r},\tag{B4}$$

with $r$ being the radius of the aerosol. For sea salt aerosol $HOBr + Cl^-$ is assumed to be limited by mass accommodation, i.e.
$\Gamma \gg \alpha$, due to high concentration of $Cl^-$ in sea salt aerosol. The reacto-diffusive length scale is

$$l_r = \sqrt{\frac{D_l}{k_{\text{HOBr}+\text{X}^-}[\text{X}^-][\text{H}^+]}},\tag{B5}$$

where $D_l = 1.4 \times 10^{-5}\,\text{cm}^2\text{s}^{-1}$ is the aqueous phase diffusion coefficient for HOBr. The listed parameters are taken from Ammann et al. (2013), and $k_{\text{HOBr}+\text{Br}^-}$ is from Beckwith et al. (1996).

**B3.2   $ClNO_3 + Br^-$**

The reactive uptake coefficient is calculated as

$$\gamma = \left(\Gamma^{-1} + \alpha^{-1}\right)^{-1},\tag{B6}$$





where the mass accomodation coefficient for $ClNO_3$ is $\alpha = 0.108$ and

$$\Gamma = \frac{4WRT\sqrt{[Br^-]D_l}}{c}, \tag{B7}$$

where $c$ is the average thermal velocity of $ClNO_3$, $D_l = 5.0 \times 10^{-6}\,cm^2s^{-1}$ is the aqueous phase diffusion coefficient for $ClNO_3$, and $W = 10^6\,\sqrt{Ms}\,bar^{-1}$.

## B3.3 $O_3 + Br^-$

The reactive uptake coefficient is calculated as

$$\gamma = \Gamma_b + \Gamma_s, \tag{B8}$$

where $\Gamma_b$ is the bulk reaction coefficient,

$$\Gamma_b = \frac{4H_{O_3}RTk_{O_3+Br^-}[Br^-]l_r f(r,l_r)}{c}, \tag{B9}$$

with $k_{O_3+Br^-} = 6.8 \times 10^8 \exp(-4450\,K/T)\,M^{-1}s^{-1}$. In the equation above $c$ is the average thermal velocity of $O_3$, and $f(l_r,r)$ is a reacto-diffusive correction factor,

$$f(l_r,r) = \coth\left(\frac{r}{l_r}\right) - \frac{l_r}{r}, \tag{B10}$$

with $r$ being the radius of the aerosol. The reacto-diffusive length scale is

$$l_r = \sqrt{\frac{D_l}{k_{O_3+Br^-}[Br^-]}}, \tag{B11}$$

where $D_l = 8.9 \times 10^{-6}\,cm^2s^{-1}$ is the aqueous phase diffusion coefficient for $O_3$.

The surface reaction coefficient is calculated as,

$$\Gamma_s = \frac{4k_s[Br^-(surf)]K_{LangC}N_{max}}{c(1 + K_{LangC}[O_3(g)])}, \tag{B12}$$

where the surface reaction rate constant is $k_s = 10^{-16}\,cm^2s^{-1}$, the equilibrium constant for $O_3$ is $K_{LangC} = 10^{-13}\,cm^3$, and the maximum number of available sites is taken as $N_{max} = 3 \times 10^{14}\,cm^{-2}$. The surface bromide concentration is estimated as,

$$[Br^-(surf)] = \min(3.41 \times 10^{14}\,cm^{-2}M^{-1}[Br^-], N_{max}). \tag{B13}$$

*Acknowledgements.* This work was funded by NERC quota studentship NE/K500987/1 with support from the NERC BACCHUS and CAST projects NE/L01291X/1, NE/J006165/1.

J. A. Schmidt acknowledges funding through a Carlsberg Foundation post-doctoral fellowship (CF14-0519)

R. Volkamer acknowledges funding from US National Science Foundation CAREER award ATM-0847793, AGS-1104104, and AGS-1452317. The involvement of the NSF-sponsored Lower Atmospheric Observing Facilities, managed and operated by the National Center for Atmospheric Research (NCAR) Earth Observing Laboratory (EOL), is acknowledged.

T. Sherwen would like to acknowledge constructive comments and input from GEOS-Chem Support Team.



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





**Figure 1.** Average annual halogen surface emission of species and column integrated fluxes for species that have fixed surface concentrations in the model ($CH_3Cl$, $CH_3Cl_2$, $CHCl_3$, $CHBr_3$) or those with vertically variable sources ($ClNO_2$ from $N_2O_5$ uptake on sea-salt and IX (X=Cl,Br) production from HOI, $lNO_2$, and $lNO_3$ uptake). Values are given in kg X $m^{-2}$ $s^{-1}$ (X=Cl,Br,I).





**Figure 1.** Continued.



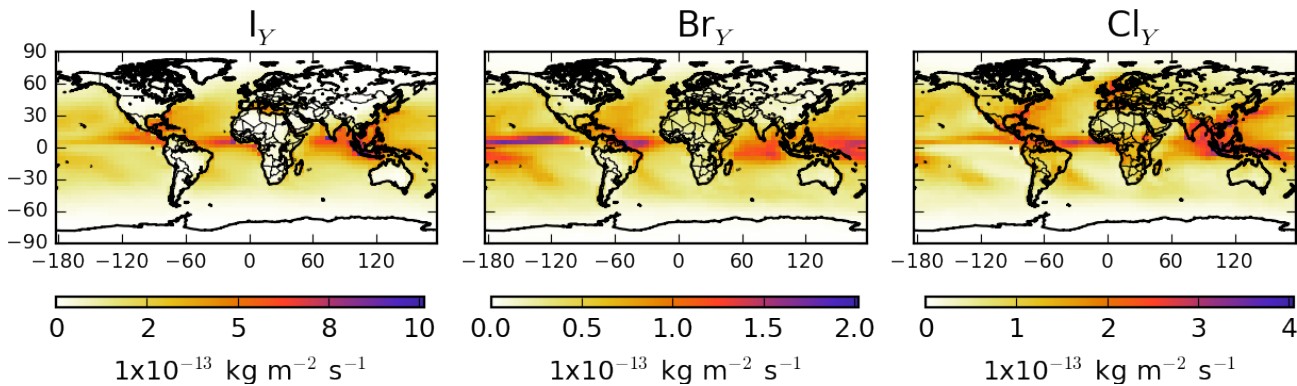

**Figure 2.** Annual global Xy (X=Cl, Br, I) deposition (defined in Footnote 1). Values are given in terms of mass of halogen deposited (kg X m$^{-2}$ s$^{-1}$, X=Cl,Br,I).



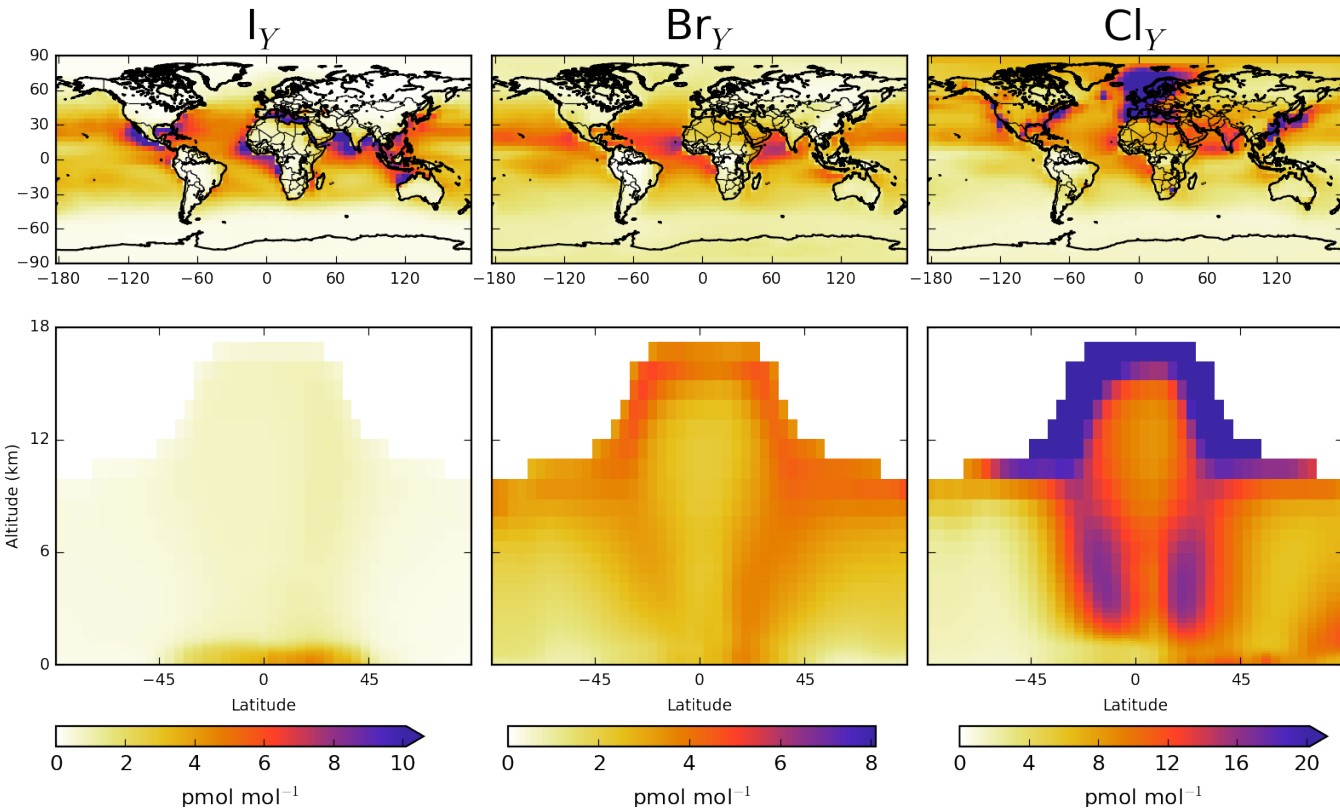

**Figure 3.** Tropospheric distribution of $Cl_Y$, $Br_Y$, and $I_Y$ (defined in Footnote 1) concentrations. Upper plots show surface and lower plots show zonal values. Only boxes that are entirely tropospheric are included in this plot. The $Cl_Y$ colourbar is capped at 20 pmol mol$^{-1}$, with a maximum plotted value of 118 pmol mol$^{-1}$ at the surface over the North Sea. The $I_Y$ colorer is capped at 10 pmol mol$^{-1}$, with a maximum plotted value of 17 pmol mol$^{-1}$ at the surface over the Red Sea.





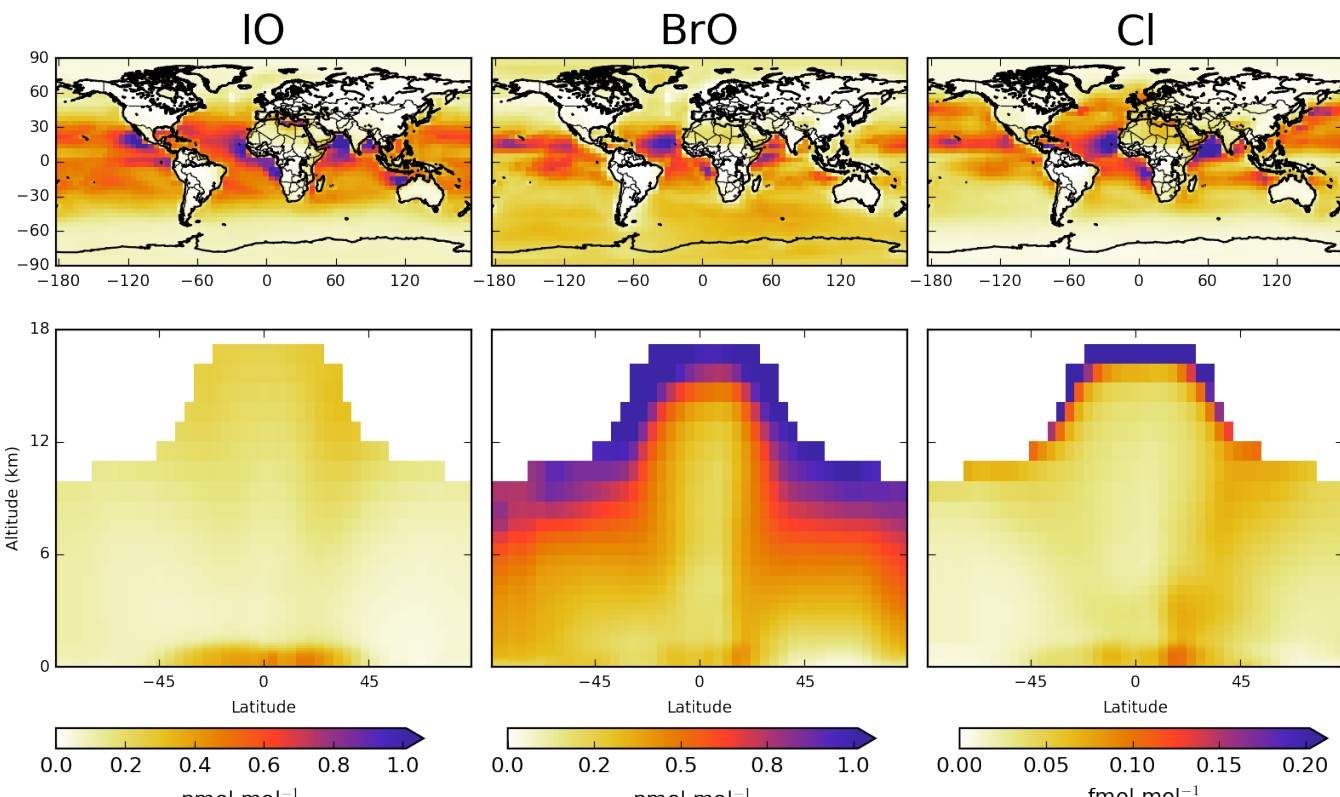

**Figure 4.** Tropospheric distribution of IO, BrO and Cl concentrations. Upper plots show surface and lower plots show zonal values. Only boxes that are entirely tropospheric are included in this plot.

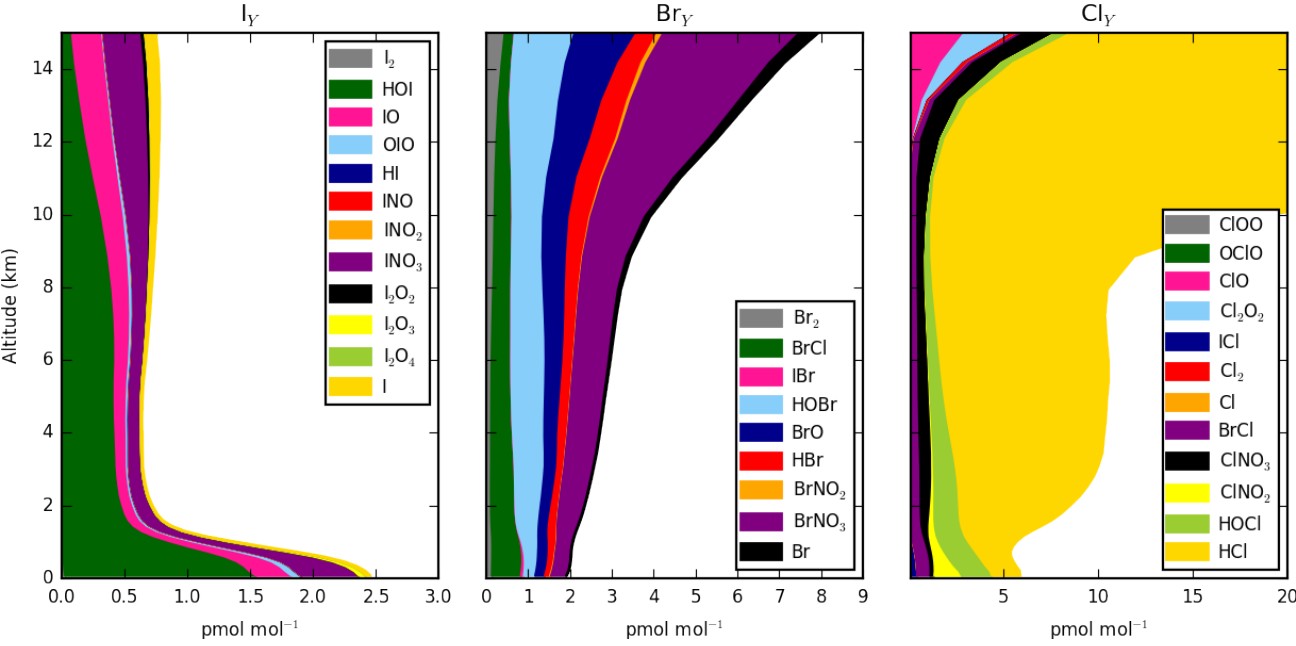

**Figure 5.** Modelled global average vertical Xy (X=Cl, Br, I) (defined in Footnote 1). Units are pmol mol$^{-1}$ of X (where X=Cl, Br, I). For Cl$_Y$ the y-axis is capped at 20 pmol mol$^{-1}$ to show speciation. A Cl$_Y$ maximum of 1062 pmol mol$^{-1}$ is found within the altitudes shown due to additional HCl contributions increasing with altitude.



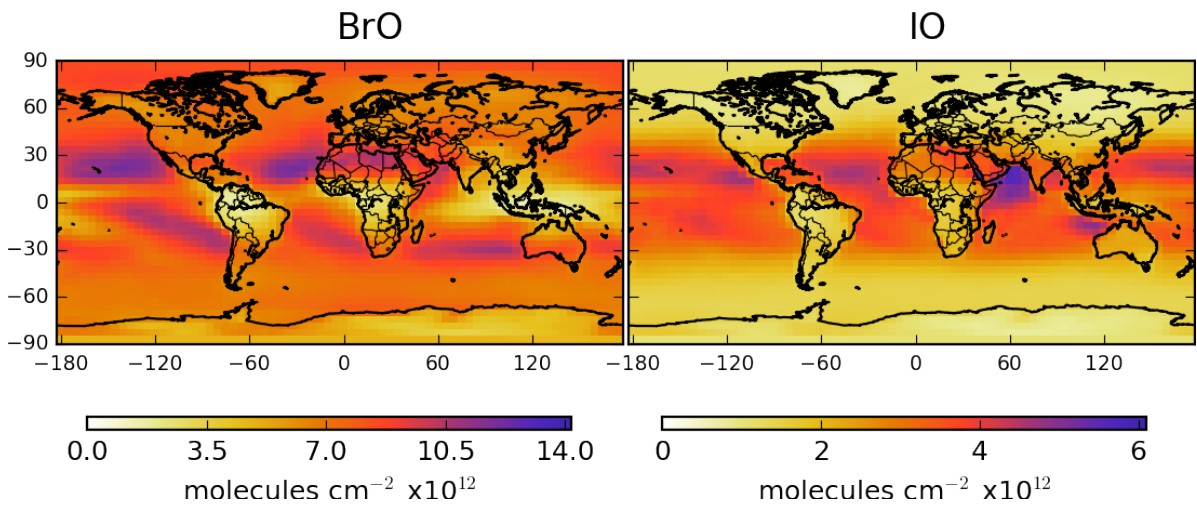

**Figure 6.** Annual mean integrated model tropospheric column for BrO and IO in molecules cm$^{-2}$.







**Figure 7.** Iodine oxide (IO) surface observations (black) by campaign compared against the simulation with halogen chemistry ("Cl+Br+I", red). Cape Verde measurements are shown against hour of day and others are shown as a function of latitude. Values are considered in 20° bins, with observations and modelled values at the same location and time (as described in section 2) shown side-by-side around the mid point of each bin. The extent of the bins is highlighted with grey dashed lines. Observations are from Cape Verde (Tropical Atlantic, Mahajan et al. 2010; Read et al. 2008), TransBrom (West Pacific, Großmann et al. 2013), the Malaspina circumnavigation (Prados-Roman et al., 2015b), HaloCAST-P (East Pacific, Mahajan et al. 2012), and TORERO ship (East Pacific, Volkamer et al. 2015). The number of data points within latitudinal bin are shown as "n". The boxplot extents give the inter-quartile range, with the median shown within the box. The whiskers give the most extreme point within 1.5 times the inter-quartile range.




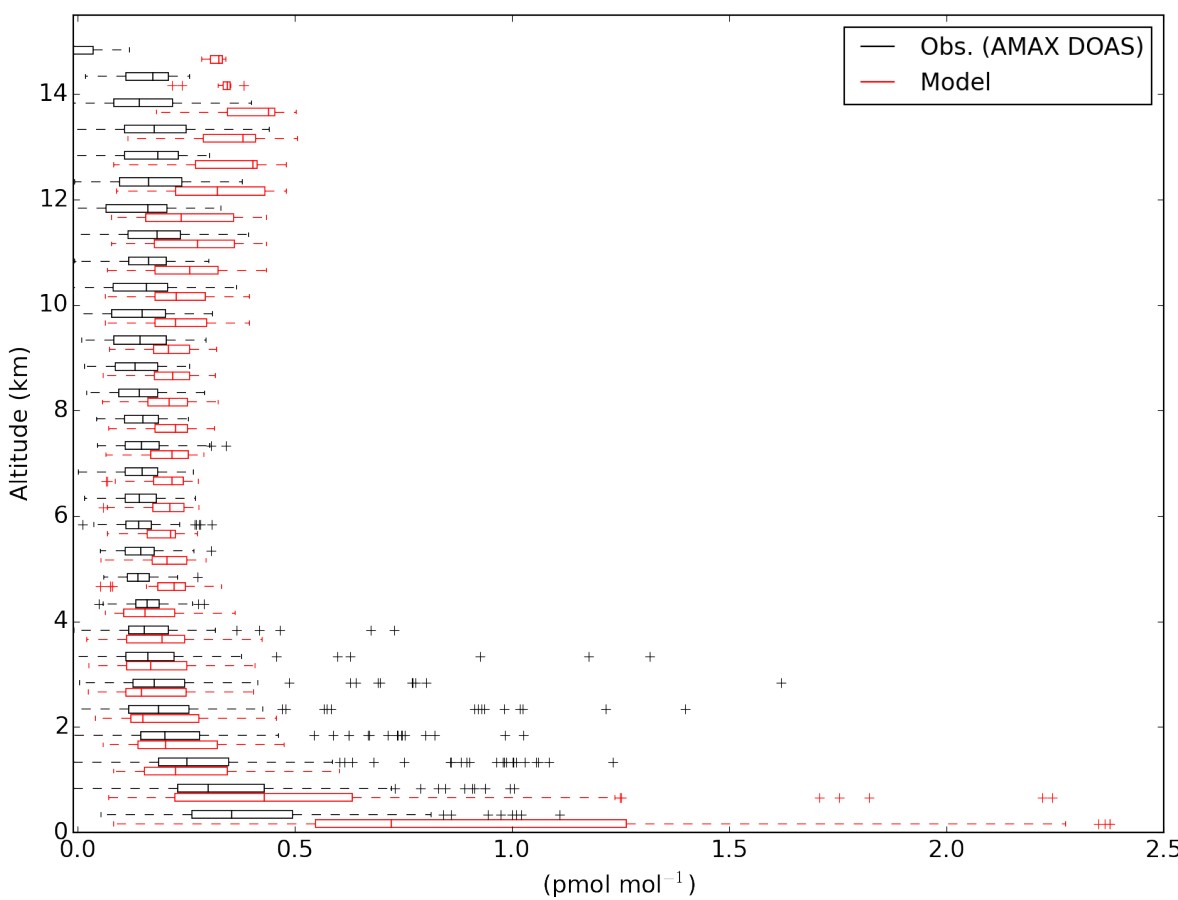

**Figure 8.** Vertical comparison of the model ("Cl+Br+I") and measured iodine oxide (IO) during TORERO aircraft campaign (Volkamer et al., 2015; Wang et al., 2015). Model and observations are in red and black respectively. Values are considered in 0.5 km bins, with observations and modelled values at the same location and time (as described in section 2) shown side-by-side around the mid point of each bin. Measurements were taken aboard the NSF/NCAR GV research aircraft by the University of Colorado airborne Multi-Axis DOAS instrument (CU AMAX-DOAS) in the eastern Pacific in January and February 2012 (Volkamer et al., 2015; Wang et al., 2015). The boxplot extents give the inter-quartile range, with the median shown within the box. The whiskers give the most extreme point within 1.5 times the inter-quartile range.





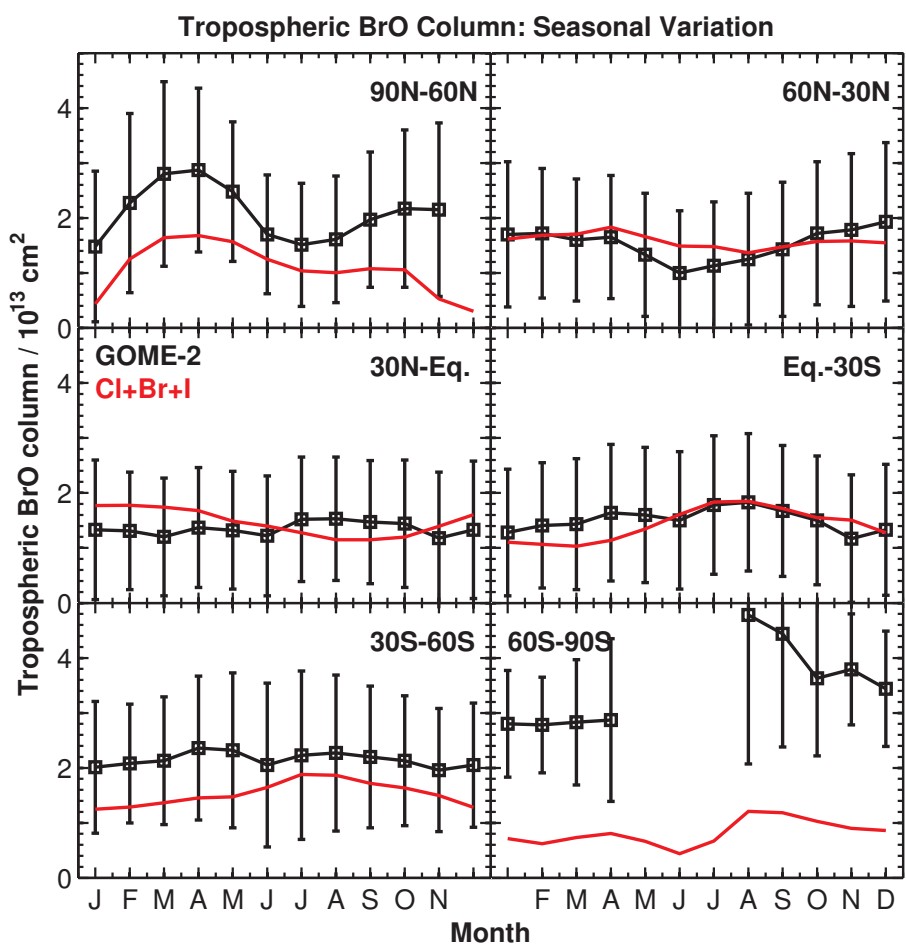

**Figure 9.** Seasonal variation of zonal mean tropospheric BrO columns in different latitudinal bands. 2007 observations from the GOME-2 satellite instrument (Theys et al., 2011) are compared to GEOS-Chem values at the GOME- 2 local overpass time (9:00-11:00).



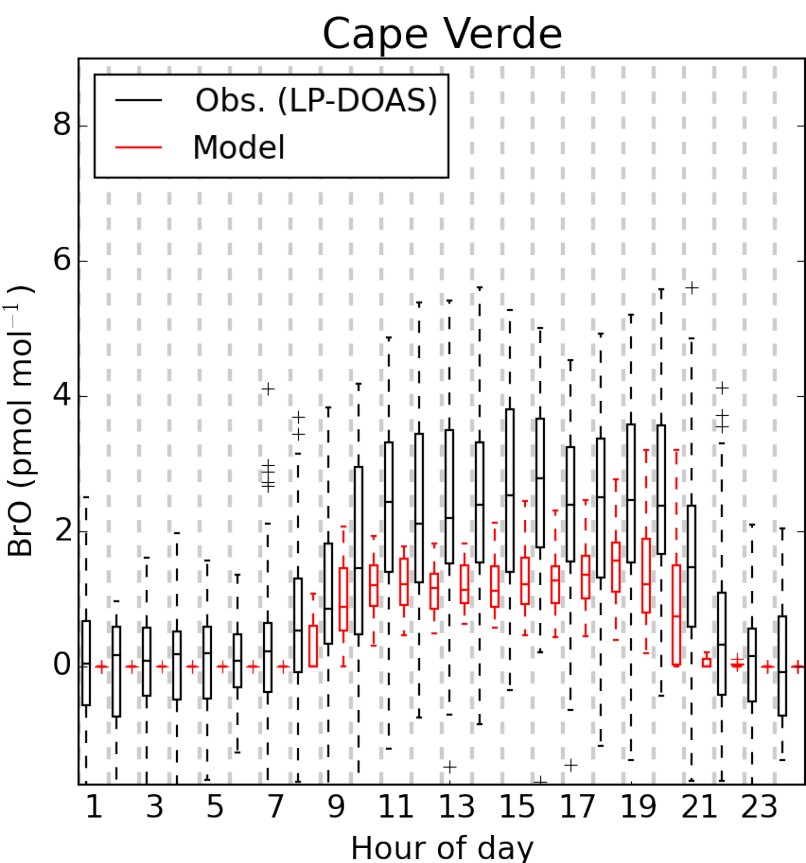

**Figure 10.** Bromine oxide (BrO) surface observations (black) at Cape Verde (Read et al., 2008; Mahajan et al., 2010) compared against the simulation with halogen chemistry ("Cl+Br+I", red). Values are binned by hour of day.





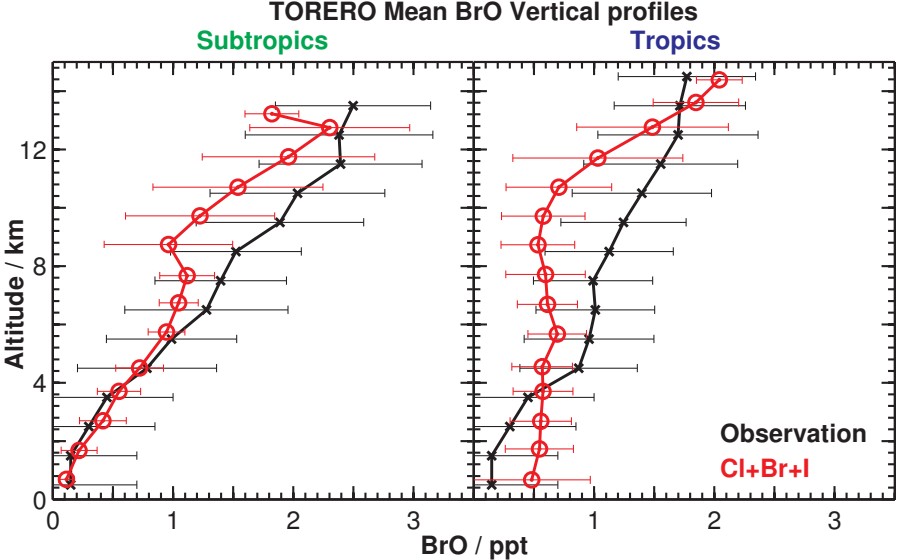

**Figure 11.** Vertical comparison of the model ("Cl+Br+I") and measured iodine oxide (BrO) during TORERO aircraft campaign (Volkamer et al., 2015; Wang et al., 2015) in the Subtropics (left) and Tropics (right).. Model and observations are in red and black, respectively. Observations and modelled values at the same location and time (as described in section 2) are shown side-by-side around the mid point of each bin. Measurements were taken aboard the NSF/NCAR GV research aircraft by the University of Colorado airborne Multi-Axis DOAS instrument (CU AMAX-DOAS) in the eastern Pacific in January and February 2012 (Volkamer et al., 2015; Wang et al., 2015). Observations below 4 km were at or below the limit of detection, which is illustrated with a dashed green line ($\sim$0.5 pmol mol$^{-1}$).



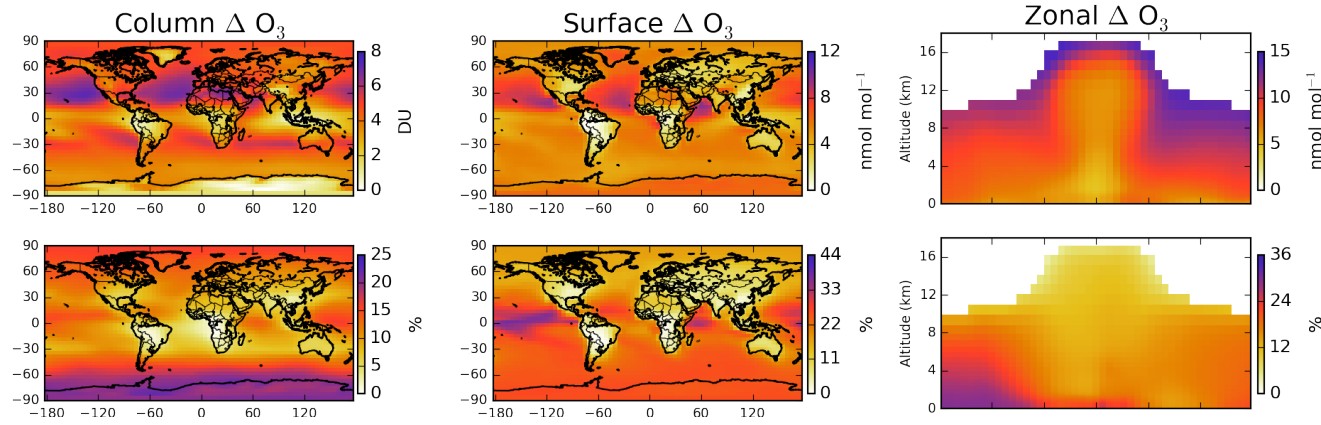

**Figure 12.** Change in tropospheric $O_3$ on inclusion of halogen chemistry. Column (left), surface (middle) and zonal (right) change are shown. Upper plots show absolute change and lower plots below give change in % terms ( ("Cl+Br+I"-"NOHAL")/"NOHAL"*100).




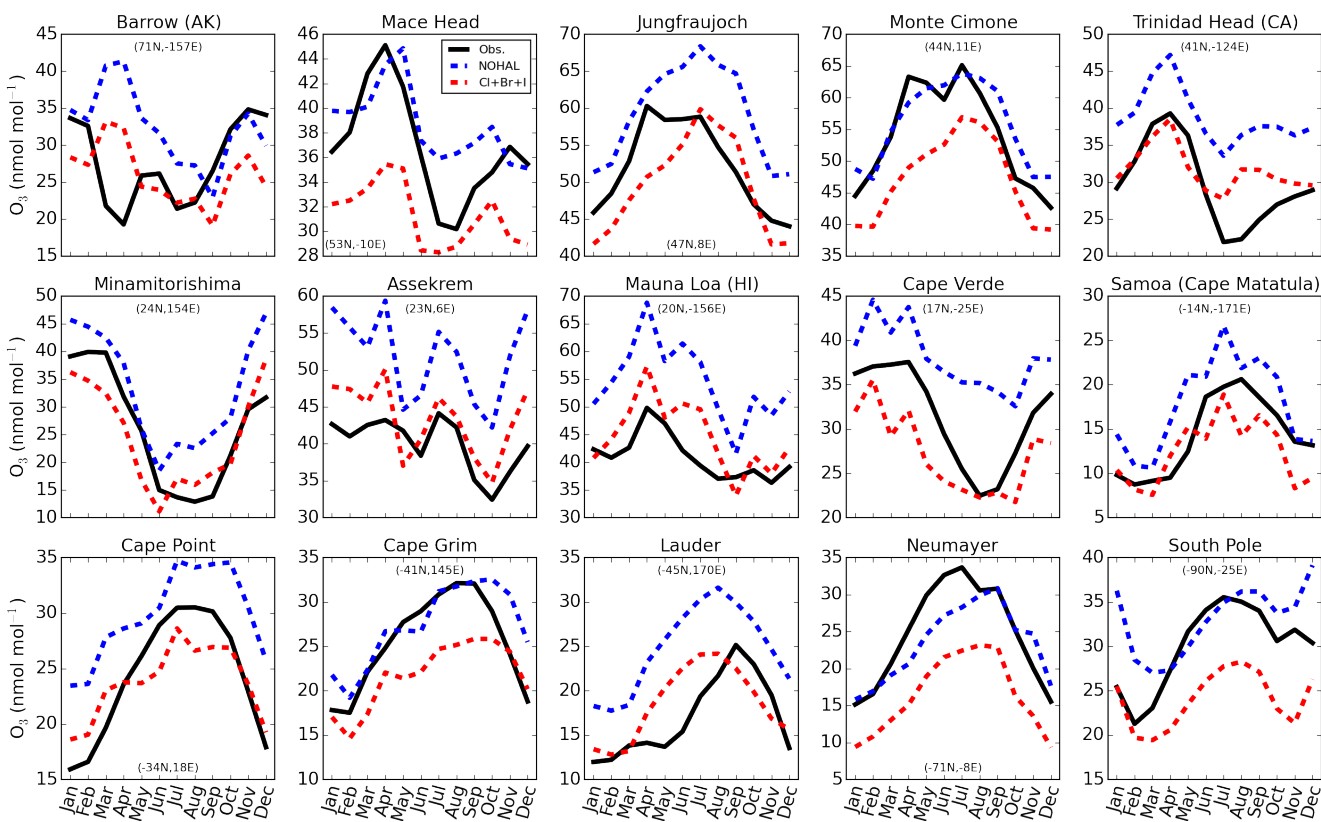

**Figure 13.** Seasonal cycle of near-surface O$_3$ at a range of Global Atmospheric Watch (GAW) sites . Observational data shown are 6 year monthly averages (2006-2012). Model data is for 2005. Data is from GAW compiled and processed as described in (Sofen et al., 2016). Blue and red lines represent simulations without halogens ("NOHAL") with halogens ("Cl+Br+I"), respectively.





**Figure 14.** Comparison between annual modelled O$_3$ profiles and sonde data (2005). Profiles shown are the annual mean of available observations from World Ozone and Ultraviolet Radiation Data Centre (WOUDC, 2014) and model data for 2005 at given locations. Blue and red lines represent simulations without halogens ("NOHAL") with halogens ("Cl+Br+I"), respectively. Observations (in black) show mean concentrations with upper and lower quartiles given by whiskers.



**Figure 15.** Global annual average tropospheric vertical odd oxygen loss ($O_X$) through different reaction routes (Photolysis, $HO_X$, $IO_X$, $BrO_X$, and $ClO_X$).





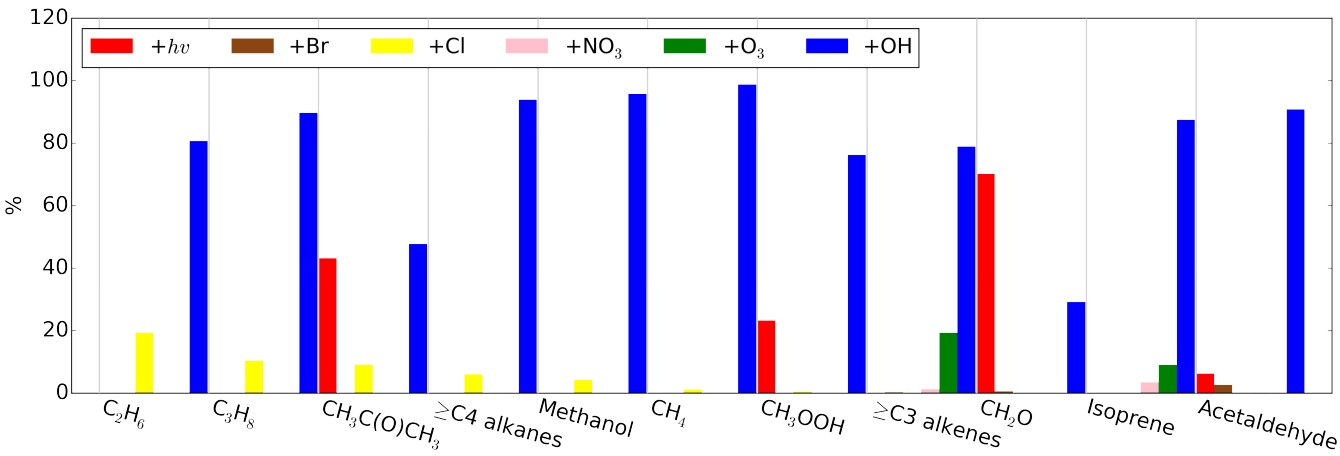

**Figure 16.** Global loss routes (+$h\nu$, +Br, +NO$_3$, +Cl, +OH) of organic compounds shown as % of total tropospheric losses.

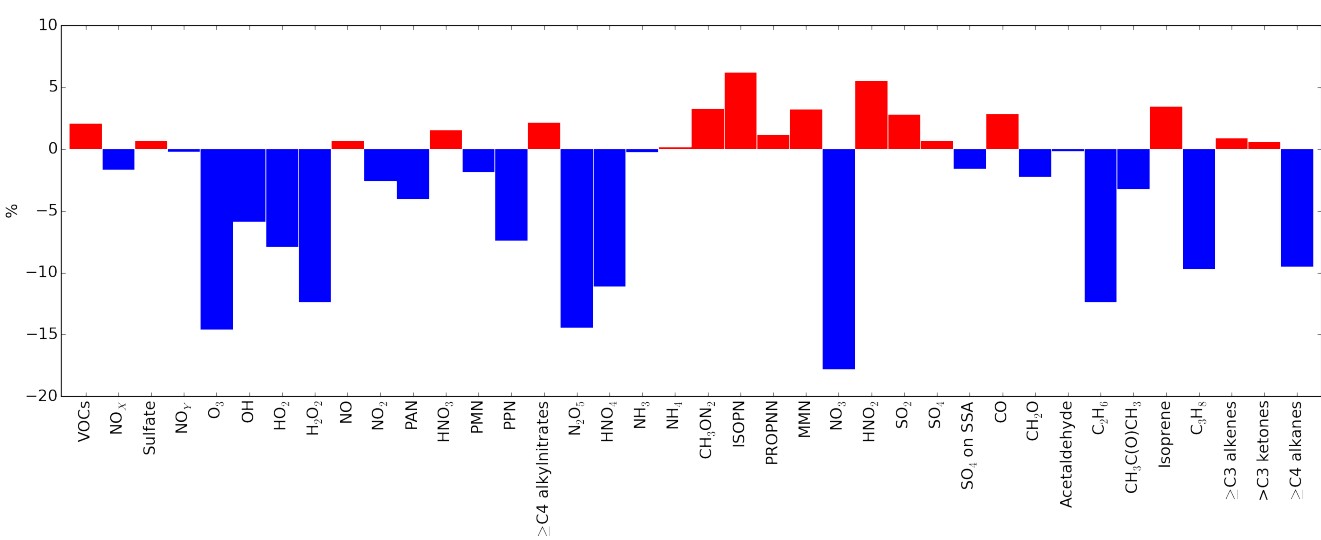

**Figure 17.** Changes in tropospheric burden of species and families on inclusion of halogens ("Cl+Br+I") compared to no halogens ("NO-HAL"). Burdens are considered in elemental terms (e.g Tg S/N/C) and species masses for OH, HO$_2$, H$_2$O$_2$ and O$_3$. The family denoted by "VOCs" in this plot is defined as the sum of carbon masses of CO, formaldehyde, acetaldehyde, ethane, acetone, isoprene, propane, ≥C4 alkanes, ≥C3 alkenes, and ≥3C ketones. Abbreviations for tracers are expanded in Footnote 3

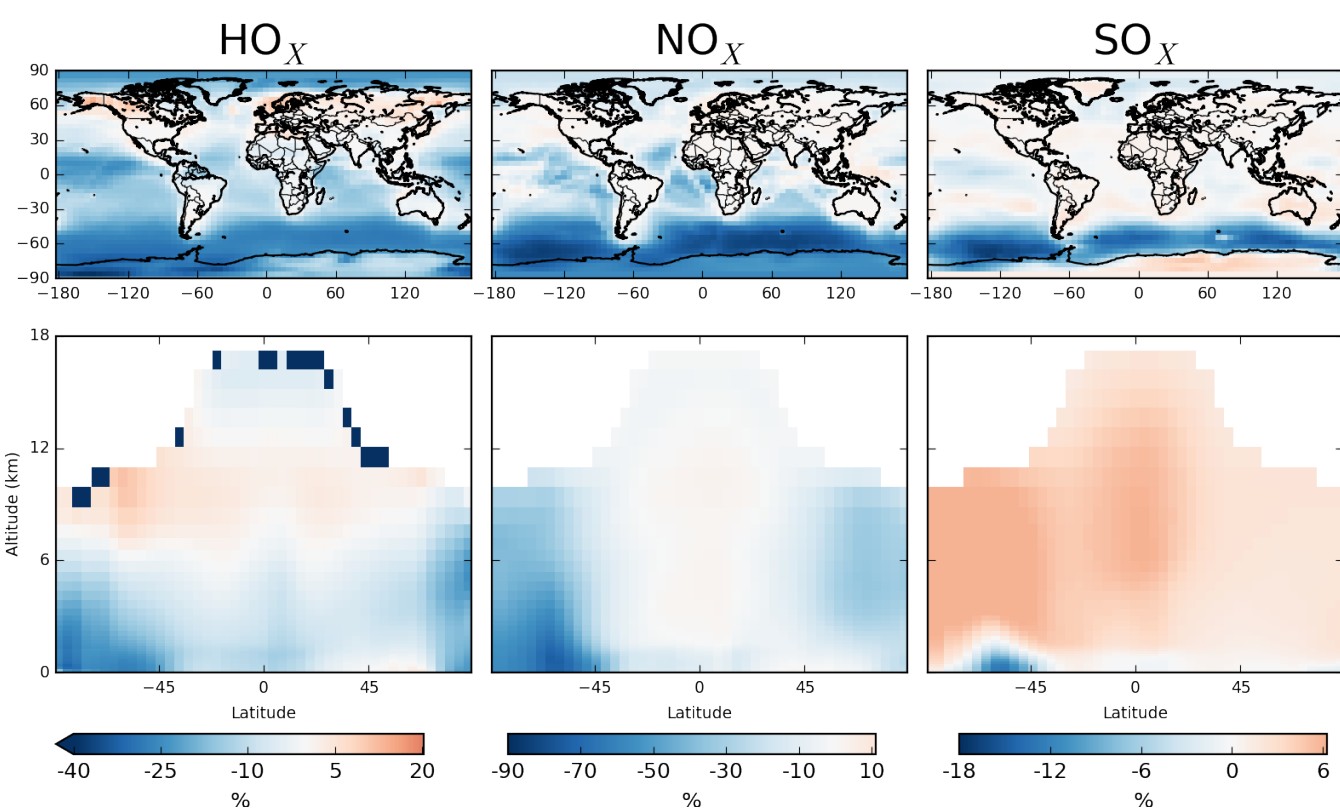

**Figure 18.** Global annual average surface and zonal change (%) in HO$_X$, NO$_X$ and SO$_X$ families (as defined in Footnote 4) on inclusion of halogens

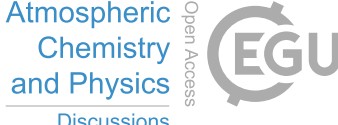

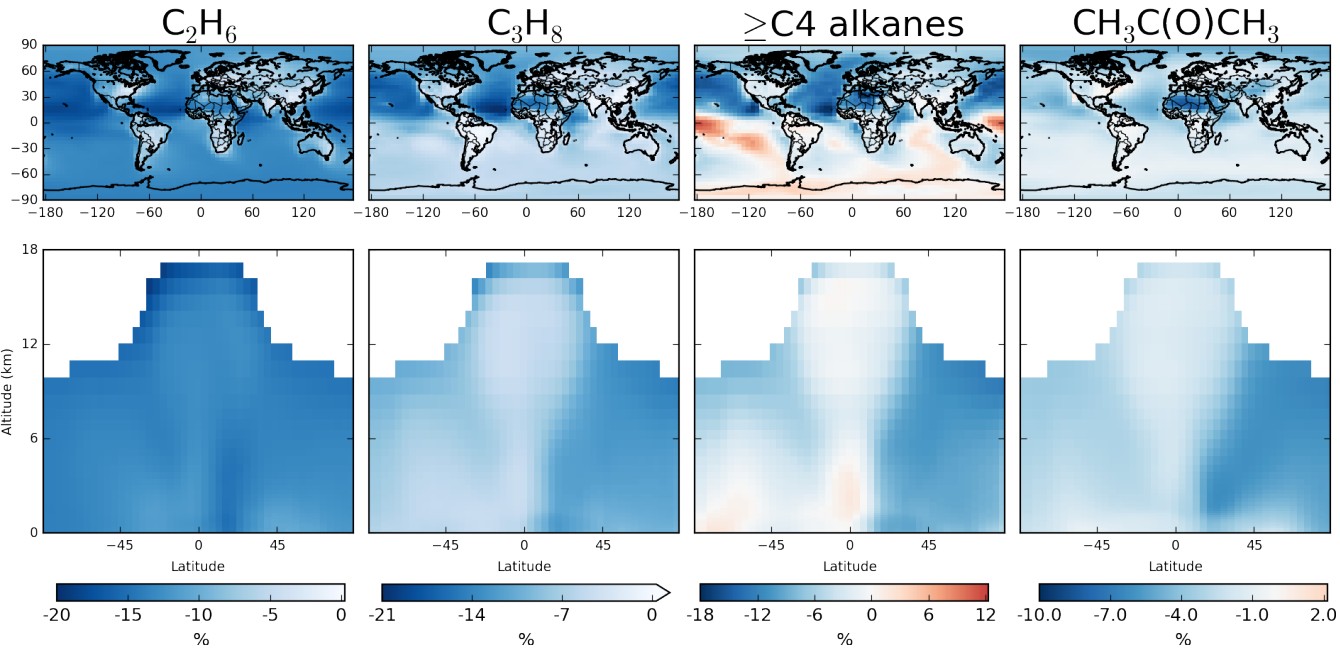

**Figure 19.** Global annual average surface and zonal change (%) in ethane ($C_2H_6$), propane ($C_3H_8$), $\geq$C4 alkanes, and acetone ($CH_3C(O)CH_3$) on inclusion of halogens. For species where all average changes are negative a continuous colourbar is used ($C_3H_8$ and $C_2H_6$) and for species where both negative and positive changes are present a divergent colourbar is used ($\geq$C4 alkanes and $CH_3C(O)CH_3$)





**Table 1.** Additional halogen reactions included in this simulation that are not described in previous work (Eastham et al., 2014; Schmidt et al., 2016; Sherwen et al., 2016). The full reaction scheme is given in the Appendix (Sections 6-9). The rate constant is calculated using a standard Arrhenius expression $Ae^{-(Ea/RT)}$

| Rxn ID | Reaction | $A$ $cm^3\ molecules^{-1}\ s^{-1}$ | $Ea/R$ K | Citation |
|---|---|---|---|---|
| M29 | $IO + ClO \rightarrow I + OClO$ | $2.59 \times 10^{-12}$ | 280 | Atkinson et al. (2007) |
| M30 | $IO + ClO \rightarrow I + Cl + O_2$ | $1.18 \times 10^{-12}$ | 280 | Atkinson et al. (2007) |
| M31 | $IO + ClO \rightarrow ICl + O_2$ | $9.40 \times 10^{-13}$ | 280 | Atkinson et al. (2007) |
| M32 | $Cl + HCOOH \rightarrow HCl + CO2 + H_2O$ | $2.00 \times 10^{-13}$ | - | Sander et al. (2011) |
| M33 | $Cl + CH_3O_2 \rightarrow ClO + CH_2O + HO_2(\bullet)$ | $1.60 \times 10^{-10}$ | - | Sander et al. (2011) |
| M34 | $Cl + CH_3OOH \rightarrow HCl + CH_3O_2$ | $5.70 \times 10^{-11}$ | - | Sander et al. (2011) |
| M35 | $Cl + C_2H_6 \rightarrow HCl + C_2H_5O_2$ | $7.20 \times 10^{-11}$ | -70 | Sander et al. (2011) |
| M36 | $Cl + C_2H_5O_2 \rightarrow ClO + HO_2 + ALD2\ (\star)$ | $7.40 \times 10^{-11}$ | - | Sander et al. (2011) |
| M37 | $Cl + EOH \rightarrow HCl + ALD2\ (\triangleleft)$ | $9.60 \times 10^{-11}$ | - | Sander et al. (2011) |
| M38 | $Cl + CH_3C(O)OH \rightarrow HCl + CH_3O_2, + CO2$ | $2.80 \times 10^{-14}$ | - | Sander et al. (2011) |
| M39 | $Cl + C_3H_8 \rightarrow HCl + A3O2$ | $7.85 \times 10^{-11}$ | -80 | Sander et al. (2011) |
| M40 | $Cl + C_3H_8 \rightarrow HCl + B3O2$ | $6.54 \times 10^{-11}$ | - | Sander et al. (2011) |
| M41 | $Cl + ACET \rightarrow HCl + ATO2$ | $7.70 \times 10^{-11}$ | -1000 | Sander et al. (2011) |
| M42 | $Cl + ISOP \rightarrow HCl + RIO2$ | $7.70 \times 10^{-11}$ | 500 | Sander et al. (2011) |
| M43 | $Cl + MOH \rightarrow HCl + CH_2O + HO_2$ | $5.50 \times 10^{-11}$ | - | Sander et al. (2011) |
| M61 | $Cl + ALK4 \rightarrow HCl + R4O2$ | $2.05 \times 10^{-10}$ | - | Atkinson et al. (2006) |
| M62 | $Br + PRPE \rightarrow HCl + PO2$ | $3.60 \times 10^{-12}$ | - | Atkinson et al. (2006) |
| M63 | $Cl + PRPE \xrightarrow{M} HCl + PO2 + M$ | $2.80 \times 10^{-10} (\$)$ | - | Atkinson et al. (2006) |
| H1 | $N_2O_5 \xrightarrow{\gamma} HNO_3 + ClNO_2 (\ominus)$ | - | - | (see table footnote) |
| H2 | $HOI \xrightarrow{\gamma} 0.85ICl + 0.15IBr*$ | - | - | (see table footnote) |
| H3 | $INO_2 \xrightarrow{\gamma} 0.85ICl + .015IBr*$ | - | - | (see table footnote) |
| H4 | $INO_3 \xrightarrow{\gamma} 0.85ICl + 0.15IBr*$ | - | - | (see table footnote) |
| P1 | $ICl \xrightarrow{h\nu} I + Cl$ | - | - | Sander et al. (2011) |
| P2 | $IBr \xrightarrow{h\nu} I + Br$ | - | - | Sander et al. (2011) |

Table footnote: ($\bullet$) Reaction from JPL, only considering the major channel ( Dale and Poulet. 1996 ) and product of $CH_3O$ reacts to form $CH_2O + HO_2$ ( $CH_3O + O_2 \rightarrow CH_2O + HO_2$). ($\star$) Only first channel from JPL considered. the 2nd channel forms a criegee ($HCl + C_2H_4O_2$ ) and therefore cannot be represented by reduced GEOS-Chem chemistry scheme.($\triangleleft$) Reaction defined by JPL and interpreted as proceeding via hydrogen abstraction, therefore the acetaldehyde product is assumed. ($\$$) K(infinity) rate given in table, K(0) rate = $4.00 \times 10^{-28}$ with Fc=0.6 as shown in Table 7. ($\ominus$) Reaction only proceeds on sea-salt aerosol. (*) Reactions which were included in previous work (Sherwen et al. (2016)), but di-halogen products have been updated split between ICl and IBr (See Section 2) and only proceed on acidic sea-salt aerosol following McFiggans et al. (2000). Acidity of aerosol is calculated as described in Alexander (2005). Abbreviations for tracers are expanded in footnote 3.



**Table 2.** Global sources of reactive tropospheric inorganic halogens. Sources with fixed concentration in the model for $Cl_Y$ ($CH_3Cl$, $CH_3Cl_2$, $CHCl_3$) and $Br_Y$ ($CHBr_3$) are shown in terms of chemical release (e.g. +Cl, +OH, +$h\nu$) and are in bold. Inclusion of chlorine and bromine organic species has been reported before in GEOS-Chem (Eastham et al., 2014; Parrella et al., 2012; Schmidt et al., 2016). $X_2$ ($I_2$) and HOX (HOI) are the inorganic ocean source from (Carpenter et al., 2013), $XNO_2$ is the source from the uptake of $N_2O_5$ on sea-salt ($ClNO_2$).

| Sources | $I_Y$ (Tg I yr$^{-1}$) | $Br_Y$ (Tg Br yr$^{-1}$) | $Cl_Y$ (Tg Cl yr$^{-1}$) |
|---|---|---|---|
| $CH_3X$ | 0.26 | **0.06** | **2.19** |
| $CH_2X_2$ | 0.33 | 0.09 | **0.59** |
| $CHX_3$ | - | 0.41 | **0.26** |
| HOX | 2.02 | - | - |
| $X_2$ | 0.14 | - | - |
| IX | - | 0.31(*) | 0.78(*) |
| $XNO_2$ | - | - | 0.66 |
| stratosphere | 0.00 | 0.06 | 0.43 |
| total source(*) | 2.75 | 0.92 | 4.90 |

(*) Note: Acid catalysed sea-salt IX (X=Cl, Br) flux only stated for $Cl_Y$ and $Br_Y$ as it does not represent a net $I_Y$ source.

**Table 3.** Comparison between modelled and observed $ClNO_2$. Concentrations are shown as the maximum and average of the daily maximum value for the observational and equivalent model time period. Sites marked as (**) are considered continental sites. The model value are taken for the nearest time-step and location within the analysis year (2005).

| Location | Lat. | Lon. | Obs. Max | Obs. Mean | "Cl+Br+I" Max | "Cl+Br+I" Mean | Reference |
|---|---|---|---|---|---|---|---|
| *Coastal* | | | | | | | |
| Pasedena, CA (2010) | 34.2 | -118.2 | 3.46 | 1.48 | 0.44 | 0.20 | Mielke et al. (2013) |
| Southern China (2012) | 22.2 | 114.3 | 2.00 | 0.31 | 0.61 | 0.18 | Tham et al. (2014) |
| Los Angeles, California (2010) | 34.1 | -118.2 | 1.83 | 0.50 | 0.44 | 0.20 | Riedel et al. (2012) |
| Houston, Texas (2006) | 30.4 | -95.4 | 1.15 | 0.80 | 0.19 | 0.04 | Osthoff et al. (2008) |
| London, UK (2012) | 51.5 | -0.2 | 0.73 | 0.23 | 0.51 | 0.17 | Bannan et al. (2015) |
| Texas (2013) | 30.4 | -95.4 | 0.14 | 0.08 | 0.19 | 0.04 | Faxon et al. (2015) |
| *Continental* | | | | | | | |
| Hessen, Germany (2011) | 50.2 | 8.5 | 0.85 | 0.20 | 0.16 | 0.02 | Phillips et al. (2012) |
| Boulder, Colorado (2009) | 40.0 | -105.3 | 0.44 | 0.14 | 0.00 | 0.00 | Thornton et al. (2010); Riedel et al. (2013) |
| Calgary, CAN (2010) | 51.1 | -114.1 | 0.24 | 0.22 | 0.02 | 0.01 | Mielke et al. (2011) |



**Table 4.** Comparison between global tropospheric $O_X$ budgets of simulations "Cl+Br+I" (with halogen chemistry) and "NOHAL" (without halogen chemistry). Recent average model values from ACCENT (Young et al., 2013) are also shown for comparison. For the $X_1O + X_2O$ halogen crossover reactions where $X_1O \neq X_2O$ we split the $O_3$ loss equally between the two routes. Values are rounded to the nearest integer value.

| | "Cl+Br+I" | "NOHAL" | ACCENT |
|---|---|---|---|
| $O_3$ burden (Tg) | 355 | 416 | $340 \pm 40$ |
| $O_X$ chemical sources (Tg yr$^{-1}$) | | | |
| NO + HO$_2$ | 3526 | 3607 | - |
| NO + CH$_3$O$_2$ | 1327 | 1316 | - |
| NO + RO$_2$ | 524 | 508 | - |
| Total chemical $O_X$ sources (PO$_X$) | 5376 | 5431 | $5110 \pm 606$ |
| $O_X$ chemical sinks (Tg yr$^{-1}$) | | | |
| $O_3 + H_2O \xrightarrow{h\nu} 2OH + O_2$ | 2102 | 2489 | - |
| $O_3 + HO_2 \rightarrow OH + O_2$ | 1136 | 1432 | - |
| $O_3 + OH \rightarrow HO_2 + O_2$ | 611 | 737 | - |
| HOBr $\xrightarrow{h\nu}$ Br + OH | 214 | - | - |
| HOBr + HCl $\rightarrow$ BrCl | 28 | - | - |
| HOBr + HBr $\rightarrow$ Br$_2$ + H$_2$O (aq. aerosol) | 13 | - | - |
| BrO + BrO $\rightarrow$ 2Br + O$_2$ | 8 | - | - |
| BrO + BrO $\rightarrow$ Br$_2$ + O$_2$ | 3 | - | - |
| BrO + OH $\rightarrow$ Br + HO$_2$ | 9 | - | - |
| IO + BrO $\rightarrow$ Br + I + O$_2$ | 9 | - | - |
| ClO + BrO $\rightarrow$ Br + ClOO/OClO | 2 | - | - |
| Other bromine $O_X$ sinks | 0 | - | - |
| Total bromine $O_X$ sinks | 284 | - | - |
| HOI $\xrightarrow{h\nu}$ I + OH | 457 | - | - |
| OIO $\xrightarrow{h\nu}$ I + O$_2$ | 125 | - | - |
| IO + BrO $\rightarrow$ Br + I + O$_2$ | 9 | - | - |
| IO + ClO $\rightarrow$ I + Cl + O$_2$/ ICl + O$_2$ | 0 | - | - |
| Other iodine $O_X$ sinks | 2 | - | - |
| Total iodine $O_X$ sinks | 593 | - | - |
| HOCl $\xrightarrow{h\nu}$ Cl + OH | 15 | - | - |
| CH$_3$O$_2$ + ClO $\rightarrow$ ClOO | 4 | - | - |
| ClO + BrO $\rightarrow$ Br + ClOO/OClO | 2 | - | - |
| ClNO$_3$ + HBr $\rightarrow$ BrCl | 1 | - | - |
| IO + ClO $\rightarrow$ I + Cl + O$_2$/ ICl + O$_2$ | 0 | - | - |
| Other chlorine $O_X$ sinks | 1 | - | - |
| Total chlorine $O_X$ sinks | 23 | - | - |
| Other $O_X$ sinks | 184 | 172 | - |
| Total chem. $O_X$ sinks (LO$_X$) | 4933 | 4829 | $4668 \pm 727$ |
| $O_3$ PO$_X$-LO$_X$ (Tg yr$^{-1}$) | 443 | 602 | $618 \pm 251$ |
| $O_3$ Dry deposition (Tg yr$^{-1}$) | 832 | 980 | $1003 \pm 200$ |
| $O_3$ Lifetime (days) | 22 | 26 | $22 \pm 2$ |
| $O_3$ STE (PO$_X$-LO$_X$-Dry dep.) (Tg yr$^{-1}$) | 389 | 378 | $552 \pm 168$ |





**Table 5.** Photolysis reactions of halogens included in scheme. Photolysis is described in Eastham et al. (2014) (ClNO$_2$, ClNO$_3$, and ClOO), Sherwen et al. (2016) (I$_2$, HOI, IO, OIO, INO, INO$_2$, INO$_3$, I$_2$O$_2$, I$_2$O$_3$, I$_2$O$_4$, CH$_3$I, CH$_2$I$_2$, CH$_2$ICl, CH$_2$IBr ), and Schmidt et al. (2016) (BrCl, Cl$_2$, ClO, HOCl, ClNO$_2$, ClNO$_3$, ClOO, Cl$_2$O$_2$, CH$_3$Cl, CH$_3$Cl$_2$, and CHCl$_3$.). As stated in Section 2, we have used the I$_2$O$_2$ cross-section for I$_2$O$_4$

| ID | Reaction | Cross-section reference |
|----|----------|------------------------|
| J1 | I$_2 \xrightarrow{h\nu}$ 2I | Sander et al. (2011) |
| J2 | HOI $\xrightarrow{h\nu}$ I + OH | Sander et al. (2011) |
| J3 | IO (+O$_2$) $\xrightarrow{h\nu}$ I (+ O$_3$) | Sander et al. (2011) |
| J4 | OIO $\xrightarrow{h\nu}$ I + O$_2$ | Sander et al. (2011) |
| J5 | INO $\xrightarrow{h\nu}$ I + NO | Sander et al. (2011) |
| J6 | INO$_2 \xrightarrow{h\nu}$ I + NO$_2$ | Sander et al. (2011) |
| J7 | INO$_3 \xrightarrow{h\nu}$ I + NO$_3$ | Sander et al. (2011) |
| J8 | I$_2$O$_2 \xrightarrow{h\nu}$ I + OIO | Gómez Martín et al. (2005), Spietz et al. (2005) |
| J9 | CH$_3$I $\xrightarrow{h\nu}$ I | Sander et al. (2011) |
| J10 | CH$_2$I$_2 \xrightarrow{h\nu}$ 2I | Sander et al. (2011) |
| J11 | CH$_2$ICl $\xrightarrow{h\nu}$ I + Cl | Sander et al. (2011) |
| J12 | CH$_2$IBr $\xrightarrow{h\nu}$ I + Br | Sander et al. (2011) |
| J13 | I$_2$O$_4 \xrightarrow{h\nu}$ 2OIO | see caption |
| J14 | I$_2$O$_3 \xrightarrow{h\nu}$ OIO + IO | Gómez Martín et al. (2005), Spietz et al. (2005) |
| J15 | CHBr$_3 \xrightarrow{h\nu}$ 3Br | Sander et al. (2011) |
| J16 | Br$_2 \xrightarrow{h\nu}$ 2Br | Sander et al. (2011) |
| J17 | BrO (+O$_2$) $\xrightarrow{h\nu}$ Br (+O$_3$) | Sander et al. (2011) |
| J18 | HOBr $\xrightarrow{h\nu}$ Br + OH | Sander et al. (2011) |
| J19 | BrNO$_2 \xrightarrow{h\nu}$ Br + NO$_2$ | Sander et al. (2011) |
| J20 | BrNO$_3 \xrightarrow{h\nu}$ Br + NO$_3$ | Sander et al. (2011) |
| J21 | BrNO$_3 \xrightarrow{h\nu}$ BrO + NO$_2$ | Sander et al. (2011) |
| J22 | CH$_2$Br$_2 \xrightarrow{h\nu}$ 2Br | Sander et al. (2011) |
| J23 | BrCl $\xrightarrow{h\nu}$ Br + Cl | Sander et al. (2011) |
| J24 | Cl$_2 \xrightarrow{h\nu}$ 2Cl | Sander et al. (2011) |
| J25 | ClO (+O$_2$) $\xrightarrow{h\nu}$ Cl (+O$_3$) | Sander et al. (2011) |
| J26 | OClO (+O$_2$) $\xrightarrow{h\nu}$ ClO (+O$_3$) | Sander et al. (2011) |
| J27 | Cl$_2$O$_2 \xrightarrow{h\nu}$ Cl + ClOO | Sander et al. (2011) |
| J28 | ClNO$_2 \xrightarrow{h\nu}$ Cl + NO$_2$ | Sander et al. (2011) |
| J29 | ClNO$_3 \xrightarrow{h\nu}$ Cl + NO$_3$ | Sander et al. (2011) |
| J30 | ClNO$_3 \xrightarrow{h\nu}$ ClO + NO$_2$ | Sander et al. (2011) |
| J31 | HOCl $\xrightarrow{h\nu}$ Cl + OH | Sander et al. (2011) |
| J32 | ClOO $\xrightarrow{h\nu}$ Cl | Sander et al. (2011) |
| J33 | CH$_3$Cl $\xrightarrow{h\nu}$ Cl + CH$_3$O$_2$, | Sander et al. (2011) |
| J34 | CH$_3$Cl$_2 \xrightarrow{h\nu}$ 2Cl | Sander et al. (2011) |



**Table 6.** Bimolecular halogen reactions included in scheme. This includes reactions from previous updates to descriptions of halogen chemistry in GEOS-Chem (Parrella et al. (2012); Eastham et al. (2014); Schmidt et al. (2016); Sherwen et al. (2016)), and those described in Section 2. These are given in the Arrhenius form with the rate equal to $A \cdot \exp\left(\frac{-Ea}{RT}\right)$. Unknown values are represented by a dash and these set to zero in the model, reducing the exponent to 1. The bi-molecular reactions with an M above the arrow represent termolecular reactions where the pressure dependence is not known or are uni-molecular decomposition reactions. Abbreviations for tracers are expanded in footnote3

| Rxn ID | Reaction | $A$ $\mathrm{cm^3\ molecules^{-1}\ s^{-1}}$ | $-Ea/R$ K | Citation |
|---|---|---|---|---|
| M1 | $I + O_3 \rightarrow IO + O_2$ | $2.10\times10^{-11}$ | -830 | Atkinson et al. (2007) |
| M2 | $I + HO_2 \rightarrow HI + O_2$ | $1.50\times10^{-11}$ | -1090 | Sander et al. (2011) |
| M3 | $I_2 + OH \rightarrow HOI + I$ | $2.10\times10^{-10}$ | – | Atkinson et al. (2007) |
| M4 | $HI + OH \rightarrow I + H_2O$ | $1.60\times10^{-11}$ | 440 | Atkinson et al. (2007) |
| M5 | $HOI + OH \rightarrow IO + H_2O$ | $5.00\times10^{-12}$ | – | Riffault et al. (2005) |
| M6 | $IO + HO_2 \rightarrow HOI + O_2$ | $1.40\times10^{-11}$ | 540 | Atkinson et al. (2007) |
| M7 | $IO + NO \rightarrow I + NO_2$ | $7.15\times10^{-12}$ | 300 | Atkinson et al. (2007) |
| M8 | $HO + CH_3I \rightarrow H_2O + I$ | $4.30\times10^{-12}$ | -1120 | Atkinson et al. (2008) |
| M9 | $INO + INO \rightarrow I_2 + 2NO$ | $8.40\times10^{-11}$ | -2620 | Atkinson et al. (2007) |
| M10 | $INO_2 + INO_2 \rightarrow I_2 + 2NO_2$ | $4.70\times10^{-12}$ | -1670 | Atkinson et al. (2007) |
| M11 | $I_2 + NO_3 \rightarrow I + INO_3$ | $1.50\times10^{-12}$ | – | Atkinson et al. (2007) |
| M12 | $INO_3 + I \rightarrow I_2 + NO_3$ | $9.10\times10^{-11}$ | -146 | Kaltsoyannis and Plane (2008) |
| M13 | $I + BrO \rightarrow IO + Br$ | $1.20\times10^{-11}$ | – | Sander et al. (2011) |
| M14 | $IO + Br \rightarrow I + BrO$ | $2.70\times10^{-11}$ | – | Bedjanian et al. (1997) |
| M15 | $IO + BrO \rightarrow Br + I + O_2$ | $3.00\times10^{-12}$ | 510 | Atkinson et al. (2007) |
| M16 | $IO + BrO \rightarrow Br + OIO$ | $1.20\times10^{-11}$ | 510 | Atkinson et al. (2007) |
| M17 | $OIO + OIO \rightarrow I_2O_4$ | $1.50\times10^{-10}$ | – | Gómez Martín et al. (2007) |
| M18 | $OIO + NO \rightarrow NO_2 + IO$ | $1.10\times10^{-12}$ | 542 | Atkinson et al. (2007) |
| M19 | $IO + IO \rightarrow I + OIO$ | $2.16\times10^{-11}$ | 180 | Atkinson et al. (2007) |
| M20 | $IO + IO \rightarrow I_2O_2$ | $3.24\times10^{-11}$ | 180 | Atkinson et al. (2007) |
| M21 | $IO + OIO \xrightarrow{M} I_2O_3$ | $1.50\times10^{-10}$ | – | Gómez Martín et al. (2007) |
| M22 | $I_2O_2 \xrightarrow{M} IO + IO$ | $1.00\times10^{12}$ | -9770 | Ordóñez et al. (2012) |
| M23 | $I_2O_2 \xrightarrow{M} OIO + I$ | $2.50\times10^{14}$ | -9770 | Ordóñez et al. (2012) |
| M24 | $I_2O_4 \xrightarrow{M} 2OIO$ | $3.80\times10^{-2}$ | – | Kaltsoyannis and Plane (2008) |
| M25 | $INO_2 \xrightarrow{M} I + NO_2$ | $9.94\times10^{17}$ | -11859 | (McFiggans et al., 2000) |
| M26 | $INO_3 \xrightarrow{M} IO + NO_2$ | $2.10\times10^{15}$ | -13670 | Kaltsoyannis and Plane (2008) |
| M27 | $IO + ClO \rightarrow I + OClO$ | $2.59\times10^{-12}$ | 280 | Atkinson et al. (2007) |
| M28 | $IO + ClO \rightarrow I + Cl + O_2$ | $1.18\times10^{-12}$ | 280 | Atkinson et al. (2007) |
| M29 | $IO + ClO \rightarrow ICl + O_2$ | $9.40\times10^{-13}$ | 280 | Atkinson et al. (2007) |
| M30 | $Cl + HCOOH \rightarrow HCl + CO2 + H_2O$ | $2.00\times10^{-13}$ | - | Sander et al. (2011) |
| M31 | $Cl + CH_3O_2 \rightarrow ClO + CH_2O + HO_2(\star)$ | $1.60\times10^{-10}$ | - | Sander et al. (2011) |
| M32 | $Cl + CH_3OOH \rightarrow HCl + CH_3O_2$ | $5.70\times10^{-11}$ | - | Sander et al. (2011) |
| M33 | $Cl + C_2H_6 \rightarrow HCl + C_2H_5O_2$ | $7.20\times10^{-11}$ | -70 | Sander et al. (2011) |
| M34 | $Cl + C_2H_5O_2 => ClO + HO_2 + ALD2 (\star)$ | $7.40\times10^{-11}$ | - | Sander et al. (2011) |
| M35 | $Cl + EOH \rightarrow HCl + ALD2 (\triangleleft)$ | $9.60\times10^{-11}$ | - | Sander et al. (2011) |
| M36 | $Cl + CH_3C(O)OH \rightarrow HCl + CH_3O_2, + CO2$ | $2.80\times10^{-14}$ | - | Sander et al. (2011) |
| M37 | $Cl + C_3H_8 \rightarrow HCl + A3O2$ | $7.85\times10^{-11}$ | -80 | Sander et al. (2011) |
| M38 | $Cl + C_3H_8 \rightarrow HCl + B3O2$ | $6.54\times10^{-11}$ | - | Sander et al. (2011) |
| M39 | $Cl + ACET \rightarrow HCl + ATO2$ | $7.70\times10^{-11}$ | -1000 | Sander et al. (2011) |
| M40 | $Cl + ISOP \rightarrow HCl + RIO2$ | $7.70\times10^{-11}$ | 500 | Sander et al. (2011) |
| M41 | $Cl + MOH \rightarrow HCl + CH_2O + HO_2$ | $5.50\times10^{-11}$ | - | Sander et al. (2011) |
| M42 | $CHBr_3 + OH \rightarrow 3Br + CO$ | $1.35\times10^{-12}$ | -600 | Sander et al. (2011) |
| M43 | $CH_2Br_2 + OH \rightarrow 2Br + CO$ | $2.00\times10^{-12}$ | -840 | Sander et al. (2011) |
| M44 | $CH_3Br + OH \rightarrow 3Br + CO$ | $2.35\times10^{-12}$ | -1300 | Sander et al. (2011) |
| M45 | $Br + O_3 \rightarrow BrO + O_2$ | $1.60\times10^{-11}$ | -780 | Sander et al. (2011) |
| M46 | $Br + CH_2O \rightarrow HO_2 + CO$ | $1.70\times10^{-11}$ | -800 | Sander et al. (2011) |
| M47 | $Br + HO_2 \rightarrow HBr + O_2$ | $4.80\times10^{-12}$ | -310 | Sander et al. (2011) |
| M48 | $Br + CH_3CHO \rightarrow CH3CO_3$ | $1.30\times10^{-11}$ | -360 | Atkinson et al. (2007) |
| M49 | $Br + (CH_3)_2CO \rightarrow CH3C(O)CH_2OO$ | $1.66\times10^{-10}$ | -7000 | King et al. (1970) |
| M50 | $Br + C_2H_6 \rightarrow C2H5OO$ | $2.36\times10^{-10}$ | -6411 | Seakins et al. (1992) |
| M51 | $Br + C_3H_8 \rightarrow C3H7OO$ | $8.77\times10^{-11}$ | -4330 | Seakins et al. (1992) |
| M52 | $Br + BrNO_3 \rightarrow Br_2 + NO_3$ | $4.90\times10^{-11}$ | 0 | Orlando and Tyndall (1996) |
| M53 | $Br + NO_3 \rightarrow BrO + NO_2$ | $1.60\times10^{-11}$ | 0 | Sander et al. (2011) |
| M54 | $HBr + OH \rightarrow Br + H_2O$ | $5.50\times10^{-12}$ | 200 | Sander et al. (2011) |
| M55 | $BrO + NO \rightarrow Br + NO_2$ | $8.80\times10^{-12}$ | 260 | Sander et al. (2011) |
| M56 | $BrO + OH \rightarrow Br + HO_2$ | $1.70\times10^{-11}$ | 250 | Sander et al. (2011) |
| M57 | $BrO + BrO \rightarrow 2Br + O_2$ | $2.40\times10^{-12}$ | 40 | Sander et al. (2011) |
| M58 | $BrO + BrO \rightarrow Br_2 + O_2$ | $2.80\times10^{-14}$ | 860 | Sander et al. (2011) |
| M59 | $BrO + HO_2 \rightarrow HOBr + O_2$ | $4.50\times10^{-12}$ | 460 | Sander et al. (2011) |
| M60 | $Br_2 + OH \rightarrow HOBr + Br$ | $2.10\times10^{-11}$ | 240 | Sander et al. (2011) |
| M61 | $Cl + ALK4 \rightarrow HCl + R4O2$ | $2.05\times10^{-10}$ | - | Atkinson et al. (2006) |
| M62 | $Cl + PRPE \rightarrow HCl + PO2$ | $3.60\times10^{-12}$ | - | Atkinson et al. (2006) |





**Table 7.** Termolecular halogen reactions included in the scheme. This includes reactions from previous updates to halogen chemistry in GEOS-Chem (Eastham et al., 2014; Parrella et al., 2012; Schmidt et al., 2016; Sherwen et al., 2016), and those detailed in section 2. The lower pressure limit rate ($k_0$) is given by: $A_0 \cdot (\frac{300}{T})^x$. The high pressure limit is given by $k_\infty$. Fc characterises the fall off curve of the reaction as described by Atkinson et al. (2007).

| Rxn ID | Reaction | $A_0$ $cm^6$ molecules$^{-2}$ s$^{-1}$ | $x$ | $k_\infty$ $cm^3$ molecules$^{-1}$ s$^{-1}$ | Fc | Citation |
|---|---|---|---|---|---|---|
| T1 | I + NO $\xrightarrow{M}$ INO | $1.80 \times 10^{-32}$ | 1 | $1.70 \times 10^{-11}$ | 0.6 | Atkinson et al. (2007) |
| T2 | I + NO$_2$ $\xrightarrow{M}$ INO$_2$ | $3.00 \times 10^{-31}$ | 1 | $6.60 \times 10^{-11}$ | 0.63 | Atkinson et al. (2007) |
| T3 | IO + NO$_2$ $\xrightarrow{M}$ INO$_3$ | $7.70 \times 10^{-31}$ | 5 | $1.60 \times 10^{-11}$ | 0.4 | Atkinson et al. (2007) |
| T4 | Br + NO$_2$ $\xrightarrow{M}$ BrNO$_2$ | $4.20 \times 10^{-31}$ | 2.4 | $2.70 \times 10^{-11}$ | 0.6 | Sander et al. (2011) |
| T5 | BrO + NO$_2$ $\xrightarrow{M}$ BrNO$_3$ | $5.20 \times 10^{-31}$ | 3.2 | $6.90 \times 10^{-12}$ | 0.6 | Sander et al. (2011) |
| T5 | BrO + NO$_2$ $\xrightarrow{M}$ BrNO$_3$ | $5.20 \times 10^{-31}$ | 3.2 | $6.90 \times 10^{-12}$ | 0.6 | Sander et al. (2011) |
| T6 | Cl + ALK4 $\xrightarrow{M}$ HCl + R4O2 | $4.00 \times 10^{-28}$ | 0 | $2.80 \times 10^{-10}$ | 0.6 | Atkinson et al. (2006) |
| T7 | Cl + O$_2$ $\xrightarrow{M}$ ClOO | $2.20 \times 10^{-33}$ | 0 | $1.80 \times 10^{-10}$(*) | 0.6 | Sander et al. (2011) |
| T8 | Cl$_2$O$_2$ $\xrightarrow{M}$ 2ClO | $9.30 \times 10^{-6}$ | 2 | $1.74 \times 10^{15}$(*) | 0.6 | Sander et al. (2011) |
| T9 | ClO + ClO $\xrightarrow{M}$ Cl$_2$O$_2$ | $1.60 \times 10^{-21}$ | 2 | $3.00 \times 10^{-12}$(*) | 0.6 | Sander et al. (2011) |
| T10 | ClO + NO$_2$ $\xrightarrow{M}$ ClNO$_3$ | $1.80 \times 10^{-31}$ | 1.9 | $1.50 \times 10^{-11}$(*) | 0.6 | Sander et al. (2011) |
| T11 | ClOO $\xrightarrow{M}$ Cl + O$_2$ | $3.30 \times 10^{-9}$ | 0 | $2.73 \times 10^{14}$(*) | 0.6 | Sander et al. (2011) |

Table footnote: . (*) $k_\infty$(T) for reactions T7-T11 have a form of $k_\infty$(T) = $k_\infty (\frac{T}{300})^{-m}$, where m = 3.1, 4.5, 4.5, 3.4 and 3.1 respectively. Abbreviations for tracers are expanded in footnote 3.





**Table 8.** Halogen multiphase reactions and reactive uptake coefficients ($\gamma$)

| ID | Reaction | Reactive uptake coefficient ($\gamma$) | Note | Reference |
|---|---|---|---|---|
| 1 | $HCl \rightarrow Cl^{-}(SSA)$ | $4.4 \times 10^{-6} \exp(2989\,K/T)$ | Sea salt only | Ammann et al. (2013) |
| 2 | $HBr \rightarrow Br^{-}(SSA)$ | $1.3 \times 10^{-8} \exp(4290\,K/T)$ | Sea salt only | Ammann et al. (2013) |
| 3 | $HI \rightarrow I(aerosol)$ | 0.1 | | |
| 4 | $ClNO_3 \rightarrow HOCl + HNO_3$ | 0.024 | Hydrolysis | Deiber et al. (2004) |
| 5 | $BrNO_3 \rightarrow HOBr + HNO_3$ | 0.02 | Hydrolysis | Deiber et al. (2004) |
| 6 | $INO_3 \rightarrow 0.85ICl + 0.15IBr + HNO_3$ | 0.01 | Sea salt only | |
| 7 | $INO_2 \rightarrow 0.85ICl + 0.15IBr + HNO_3$ | 0.02 | Sea salt only | |
| 8 | $HOBr + Cl^{-}(aq) \rightarrow BrCl$ | See text | | Ammann et al. (2013) |
| 9 | $HOBr + Br^{-}(aq) \rightarrow Br_2$ | See text | | Ammann et al. (2013) |
| 10 | $HOI \rightarrow 0.85ICl + 0.15IBr$ | 0.01 | Sea salt only | |
| 11 | $ClNO_3 + Br^{-}(aq) \rightarrow BrCl + HNO_3$ | See text | | Ammann et al. (2013) |
| 12 | $O_3 + Br^{-}(aq) \rightarrow HOBr$ | See text | | Ammann et al. (2013) |
| 13 | $I_2O_2 \rightarrow I(aerosol)$ | 0.02 | | |
| 14 | $I_2O_3 \rightarrow I(aerosol)$ | 0.02 | | |
| 15 | $I_2O_4 \rightarrow I(aerosol)$ | 0.02 | | |





**Table 9.** Henry's law coefficients and molar heats of formation of iodine species. Where Henry's law constant equals infinity a very large values is used within the model ($1 \times 10^{20}$ M atm$^{-1}$). The $INO_2$ Henry's law constant is assumed equal to that of $BrNO_3$, from Sander (2015), by analogy. For $I_2O_X$ ($x = 2, 3, 4$) a Henry's law constant of infinity is assumed by analogy with $INO_3$. ($*$) Effective Henry's law of HX is calculated for acid conditions through $K_H^*(T) = K_H(T) \times (1 + \frac{K_a}{[H^+]})$.

| Species | Henry's Law Constant (H) at 298K M atm$^{-1}$ | Reference | $\frac{d(\ln H)}{d(1/T)}$ K | Reference |
|---|---|---|---|---|
| HOBr | $6.1 \times 10^3$ | Frenzel et al. (1998) | $6.01 \times 10^3$ | McGrath and Rowland (1994) |
| HBr(*) | $7.1 \times 10^{13}$ | Frenzel et al. (1998) | $1.02 \times 10^4$ | Schweitzer et al. (2000) |
| $BrNO_2$ | 0.3 | Frenzel et al. (1998) | - | - |
| $BrNO_3$ | $\infty$ | Sander (2015) | - | - |
| $Br_2$ | 0.76 | Dean (1992) | $3.72 \times 10^3$ | Dean (1992) |
| HOCl | $6.5 \times 10^3$ | Sander (2015) | $5.9 \times 10^3$ | Sander (2015) |
| HCl(*) | $7.1 \times 10^{15}$ | Sander (2015) | $5.9 \times 10^3$ | Sander (2015) |
| $ClNO_3$ | $\infty$ | Sander (2015) | - | - |
| BrCl | 0.97 | Sander (2015) | - | - |
| ICl | $1.11 \times 10^2$ | Sander (2015) | $2.11 \times 10^3$ | Sander et al. (2006) |
| IBr | $2.43 \times 10^1$ | Sander (2015) | $4.92 \times 10^3$ | Sander et al. (2006) |
| HOI | $1.53 \times 10^4$ | Sander (2015) | $8.37 \times 10^3$ | Sander et al. (2006) |
| HI (*) | $7.43 \times 10^{13}$ | Sander (2015) | $3.19 \times 10^3$ | Sander et al. (2006) |
| $INO_3$ | $\infty$ | Vogt et al. (1999) | $3.98 \times 10^4$ | Kaltsoyannis and Plane (2008) |
| $I_2O_2$ | $\infty$ | see caption text | $1.89 \times 10^4$ | Kaltsoyannis and Plane (2008) |
| $I_2$ | 2.63 | Sander (2015) | $7.51 \times 10^3$ | Sander et al. (2006) |
| $INO_2$ | 0.3 | see caption text | $7.24 \times 10^3$ | Sander et al. (2006) |
| $I_2O_3$ | $\infty$ | see caption text | $7.70 \times 10^3$ | Kaltsoyannis and Plane (2008) |
| $I_2O_4$ | $\infty$ | see caption text | $1.34 \times 10^4$ | Kaltsoyannis and Plane (2008) |





**Table 10.** Tropospheric burden of species and families with ("Cl+Br+I") and without halogens ("NOHAL"), and % change. Burdens are considered in elemental terms ( e.g Gg S/N/C) and species masses for OH, $HO_2$, $H_2O_2$ and $O_3$ Families are defined in footnote 3

| | "NOHAL" | "Cl+Br+I" | % $\Delta$ |
|---|---|---|---|
| $NO_3$ | 1.23 | 1.23 | -17.8 |
| $O_3$ | 415843.25 | 355123.69 | -14.6 |
| $N_2O_5$ | 9.38 | 8.02 | -14.5 |
| $H_2O_2$ | 3229.09 | 2828.80 | -12.4 |
| $C_2H_6$ | 3258.84 | 2855.31 | -12.4 |
| $HNO_4$ | 19.84 | 17.63 | -11.1 |
| $C_3H_8$ | 609.76 | 550.68 | -9.7 |
| $\geq$C4 alkanes | 488.35 | 441.96 | -9.5 |
| $HO_2$ | 27.55 | 25.37 | -7.9 |
| PPN | 15.82 | 14.65 | -7.4 |
| PAN | 202.89 | 194.70 | -4.0 |
| $CH_3C(O)CH_3$ | 7533.51 | 7289.92 | -3.2 |
| OH | 0.28 | 0.27 | -2.9 |
| $NO_2$ | 123.53 | 120.35 | -2.6 |
| $CH_2O$ | 389.55 | 380.88 | -2.2 |
| PMN | 0.68 | 0.67 | -1.8 |
| $NO_X$ | 171.01 | 168.15 | -1.7 |
| $SO_4$ on SSA | 1.97 | 1.94 | -1.6 |
| $NH_3$ | 126.61 | 126.28 | -0.3 |
| $NO_Y$ | 1374.56 | 1371.59 | -0.2 |
| Acetaldehyde | 184.93 | 184.59 | -0.2 |
| $NH_4$ | 270.93 | 271.43 | 0.2 |
| >C3 ketones | 186.99 | 188.11 | 0.6 |
| $SO_X$ | 398.98 | 401.59 | 0.7 |
| $SO_4$ | 397.01 | 399.65 | 0.7 |
| NO | 47.48 | 47.80 | 0.7 |
| $\geq$C3 alkenes | 97.93 | 98.79 | 0.9 |
| PROPNN | 7.46 | 7.55 | 1.1 |
| $HNO_3$ | 463.49 | 470.69 | 1.6 |
| VOCs | 148193.29 | 151283.71 | 2.1 |
| $\geq$C4 alkylnitrates | 64.60 | 65.99 | 2.2 |
| $SO_2$ | 286.11 | 294.17 | 2.8 |
| CO | 134654.88 | 138477.76 | 2.8 |
| MMN | 3.15 | 3.26 | 3.2 |
| $CH_3NO_2$ | 13.80 | 14.25 | 3.3 |
| Isoprene | 788.55 | 815.73 | 3.4 |
| $HNO_2$ | 2.76 | 2.92 | 5.5 |
| ISOPN | 0.65 | 0.69 | 6.2 |

Abbreviations for tracers are expanded in footnote 3.