# Peer review of "Global impacts of tropospheric halogens (Cl, Br, I) on oxidants and composition in GEOS-Chem"

_Atmospheric Chemistry and Physics, 2016_

## Referee Comment (RC1) · Anonymous Referee #1 · 3 Jul 2016

This manuscript presents a comprehensive model study of halogen chemistry; using an updated version of the global model GEOS-Chem, the authors examine the impact of halogens on the composition of the troposphere. It is a very interesting paper and it adds to the growing corpus of studies that try to assess the global impact of halogen chemistry. The manuscript is well written and the material clearly presented and discussed. I recommend publication in Atmos. Chem. Phys., with minor corrections.

GENERAL COMMENTS

In the introduction it should be noted that some halogen chemistry actually leads to increased O3 formation, due to increased oxidation of VOC and recycling of NOx. Especially since ClNO2 chemistry is highlighted later in the paper.

A few clarifications about the chemical mechanism are needed. In particular: is nucleation included for IxOy species? Or are the only losses for these species photolysis and heterogeneous uptake? Release of Cl and Br is described as only via uptake of N2O5 (page 4, line 30) which seems to contrast with the description of acid catalysed release described earlier in the same section (page 3, line 31). This (apparent?) contradiction should be clarified.

The importance of aqueous-phase chemistry is briefly mentioned in Section 3.4.3 as a possible explanation for the disagreement between modelled and measured chlorine, but is probably an issue for bromine, and maybe iodine, as well. Given that this is likely the main uncertainty in the model (with regard to halogen chemistry) more discussion seems warranted.

The section on the impact of halogens on ozone concentration should be expanded. While it is true that halogens generally improve the agreement with ozone, this is not always the case (eg, Mace Head, Mont Cimone, Neumayer in Figure 12, Lindenberg, Marambio in Fig 13). It is also quite clear that the model often fails to reproduce ozone at higher altitudes. These discrepancies should be discussed.

MINOR COMMENTS

It may not be clear to everybody where the measurements were taken. I suggest either a map indicating the location of the sites and of the campaigns or an expansion of Table 3 to include all the measurements used in the paper.

Section 4.2: is HO2 increased in the model with halogen chemistry? If so by how much.

Page 8, Line 2: higher

Page 8, Line 12: dominated

Page 10, Line 27: suggested

---

## Referee Comment (RC2) · Anonymous Referee #2 · 18 Jul 2016

This manuscript describes modeling of the global impacts of tropospheric halogens on oxidants (ozone and its photoproducts). The manuscript is well written and expands upon prior modeling work that examined one or two halogens by including all three atmospherically relevant halogens (Cl, Br, and I). The coupling of these species is of interest because cross reactions between the halogens could have significant impacts on the chemistry. The results of the modeling are compared to available observations. The paper nicely summarizes the results of the modeling efforts in figures and tables. The paper is appropriate for ACP and I recommend publication following minor revisions.

Minor comments:

On page 3, near the bottom, the photolysis of I2Ox species is discussed. The section

is not very clear with regard to "recent work". This phrase seems to refer to work other than the present ACPD paper. If so, please indicate what "recent work" is and where the "unpublished spectrum" is from.

On page 6, the discussion of general lifetimes reads well. It might be valuable to add a bit more detail on the relative XOx lifetimes. Specifically, which reaction is the major control on the XOx lifetime would be of interest. The lifetime variation (short for IOx, longer for BrOx, and then very short for ClOx) would also be interesting to be discussed in terms of chemical principles.

Page 7, line 10, Tropospheric repeats twice int he same sentence.

Page 8, line 12, hihjer misspelled

Page 8, line 23, "is dominate" needs rewording

Page 9, line 14, "at the surface concentrations" maybe "at" is the wrong word?

Page 9, line 26, With regard to ozone as a greenhouse gas, it seems like a discusion of the free tropospheric loss of ozone should be put in the context of the altitude range where ozone is greenhouse active.

Page 10, line 5, it is interesting to note that coupling of halogens (cross reactions) appear unimportant. In some literature, they point to fast rates of cross reactions, but I think the cross rates differ between measurements / evaluations; could this be discussed more fully.
* * *

---

## Author Comment (AC1) · 12 Sep 2016

Response to interactive comments on "Global impacts of tropospheric halogens (Cl, Br, I) on oxidants and composition in GEOS-Chem" by T. Sherwen et al.

General response from authors:

We thank anonymous reviewers #1 and #2 for their positive reviews and constructive comments on our paper. We have updated the manuscript following these comments and addressed all points raised. We feel that the reviewers have improved our manuscript and are grateful for their time and contributions.

Two errors in the code have been identified following submission to ACPD (The calculation of cloud surface area and a typographic error in the representation of a bromine

and VOC reaction). The conclusions of the paper are unaffected, but the magnitude of the impacts of halogen chemistry has increased slightly. For instance, the reported decrease in tropospheric ozone burden has increased (18.6% instead of ∼15 %) and tropospheric OH has also decreased (8.2 % instead of 6.5 %). We have also extended the definition of the chemical lifetime of Cly and Bry to include cloud processes which was not included in the previous version. The numbers and figures throughout the paper have been updated accordingly.

In addition to reviewer's comments, a few minor updates were made. The formatting of subscripts for chemical species (e.g. Cly, HOx) has been updated. Figures have updated with improved formatting and any repetition of labeling were removed. Finally, a link has been made in the introduction to a companion paper, which considers halogen effects in the preindustrial atmosphere.

Anonymous Referee #1:

This manuscript presents a comprehensive model study of halogen chemistry; using an updated version of the global model GEOS-Chem, the authors examine the impact of halogens on the composition of the troposphere. It is a very interesting paper and it adds to the growing corpus of studies that try to assess the global impact of halogen chemistry. The manuscript is well written and the material clearly presented and discussed. I recommend publication in Atmos. Chem. Phys., with minor corrections.

We thank reviewer #1 for the positive comments about our manuscript and we respond to the minor corrections raised below.

GENERAL COMMENTS In the introduction it should be noted that some halogen chemistry actually leads to increased O3 formation, due to increased oxidation of VOC and recycling of NOx. Especially since ClNO2 chemistry is highlighted later in the paper.

This content has been added to the introduction.

A few clarifications about the chemical mechanism are needed. In particular: is nucleation included for IxOy species? Or are the only losses for these species photolysis and heterogeneous uptake? Release of Cl and Br is described as only via uptake of N2O5 (page 4, line 30) which seems to contrast with the description of acid catalysed release described earlier in the same section (page 3, line 31). This (apparent?) contradiction should be clarified.

The manuscript has been updated to improve the clarity of the explanation of which processed are included.

The importance of aqueous-phase chemistry is briefly mentioned in Section 3.4.3 as a possible explanation for the disagreement between modelled and measured chlorine, but is probably an issue for bromine, and maybe iodine, as well. Given that this is likely the main uncertainty in the model (with regard to halogen chemistry) more discussion seems warranted.

A referenced sentence has been added to conclusions to highlight these uncertainties and to direct readers to reviews where this discussion has already happened in depth (Saiz-Lopez et al. 2012b, Simpson et al. 2015) and modeling work that has considered this (Sherwen et al. 2016a, Schmidt et al. 2016)

The section on the impact of halogens on ozone concentration should be expanded. While it is true that halogens generally improve the agreement with ozone, this is not always the case (eg, Mace Head, Mont Cimone, Neumayer in Figure 12, Lindenberg, Marambio in Fig 13). It is also quite clear that the model often fails to reproduce ozone at higher altitudes. These discrepancies should be discussed.

We accept the model does not completely reproduce the observations, but would argue that it does a reasonable job considering the complexities of O3 sources, chemistry, and transport. There are two sonde sites in Fig. 13 that are now outside the quartiles of the observations. To make the comparison comparably fair to the sonde comparison, the surface plot has been updated to display the 5th to 95th percentile

of the observations. Adding these observations highlights that for the majority of sites there is a significant improvement on inclusions of halogens. However, the simulation of ozone clearly degrades at Neumayer and the South pole and a statement has been added to highlight this.

MINOR COMMENTS It may not be clear to everybody where the measurements were taken. I suggest either a map indicating the location of the sites and of the campaigns or an expansion of Table 3 to include all the measurements used in the paper.

A figures has been added to show locations Ozone measurements and another to show locations of halogen measurements.

Section 4.2: is HO2 increased in the model with halogen chemistry? If so by how much.

As HOx is mainly consists of HO2, the numbers are essentially the same as for HOx and therefore are not included separately. Both HOx and HO2 decrease, as seen in that section for HOx and in Figure 17 for HO2.

Page 8, Line 2: higher

Updated.

Page 8, Line 12: dominated

Updated.

Page 10, Line 27: suggested

Updated.

Anonymous Referee #2

This manuscript describes modeling of the global impacts of tropospheric halogens on oxidants (ozone and its photoproducts). The manuscript is well written and expands upon prior modeling work that examined one or two halogens by including all three

atmospherically relevant halogens (Cl, Br, and I). The coupling of these species is of interest because cross reactions between the halogens could have significant impacts on the chemistry. The results of the modeling are compared to available observations. The paper nicely summarizes the results of the modeling efforts in figures and tables. The paper is appropriate for ACP and I recommend publication following minor revisions.

We thank reviewer #2 for the positive comments about our manuscript and we respond to the minor corrections raised below.

Minor comments: On page 3, near the bottom, the photolysis of I2Ox species is discussed. The section is not very clear with regard to "recent work". This phrase seems to refer to work other than the present ACPD paper. If so, please indicate what "recent work" is and where the "unpublished spectrum" is from.

The citation for this is Saiz-Lopez et al. (2014), which has been moved within the sentence to make the link clearer.

On page 6, the discussion of general lifetimes reads well. It might be valuable to add a bit more detail on the relative XOx lifetimes. Specifically, which reaction is the major control on the XOx lifetime would be of interest. The lifetime variation (short for IOx, longer for BrOx, and then very short for ClOx) would also be interesting to be discussed in terms of chemical principles.

The discussion of XOx lifetime has been expanded in the manuscript as requested by the reviewer. Elsewhere the lifetimes and controls on lifetime of iodine (Sherwen et al. 2016a) and bromine (Parrella et al. 2012, Schmidt et al. 2016) have previously been discussed.

Page 7, line 10, Tropospheric repeats twice int he same sentence.

Updated.

Page 8, line 12, hihjer misspelled

Updated.

Page 8, line 23, "is dominate" needs rewording

Updated.

Page 9, line 14, "at the surface concentrations" maybe "at" is the wrong word?

Updated.

Page 9, line 26, With regard to ozone as a greenhouse gas, it seems like a discussion of the free tropospheric loss of ozone should be put in the context of the altitude range where ozone is greenhouse active.

A sentence has been added to this effect and a link made to the companion paper (Sherwen et al 2016c), which explores impact of tropospheric O3 change in more detail and in terms of climatic implications from preindustrial to present day.

Page 10, line 5, it is interesting to note that coupling of halogens (cross reactions) appear unimportant. In some literature, they point to fast rates of cross reactions, but I think the cross rates differ between measurements / evaluations; could this be discussed more fully.

A sentence has been added to the manuscript to explain the difference between previous studies (in localized regions with high XO concentrations) and the global picture considered here.